# Inverse modelling of European CH₄ emissions during 2006-2012 using different inverse models and reassessed atmospheric observations

Peter Bergamaschi[1], Ute Karstens[2,3], Alistair J. Manning[4], Marielle Saunois[5], Aki Tsuruta[6], Antoine Berchet[5,7], Alexander T. Vermeulen[3,8], Tim Arnold[4,9,10], Greet Janssens-Maenhout[1], Samuel Hammer[11], Ingeborg Levin[11], Martina Schmidt[11], Michel Ramonet[5], Morgan Lopez[5], Jost Lavric[2], Tuula Aalto[6], Huilin Chen[12,13], Dietrich G. Feist[2], Christoph Gerbig[2], László Haszpra[14,15], Ove Hermansen[16], Giovanni Manca[1], John Moncrieff[10], Frank Meinhardt[17], Jaroslaw Necki[18], Michal Galkowski[18], Simon O'Doherty[19], Nina Paramonova[20], Hubertus A. Scheeren[12], Martin Steinbacher[7], and Ed Dlugokencky[21]

[1] European Commission Joint Research Centre, Ispra (Va), Italy
[2] Max Planck Institute for Biogeochemistry, Jena, Germany
[3] ICOS Carbon Portal, ICOS ERIC, University of Lund, Sweden
[4] Met Office Exeter, Devon, UK
[5] Laboratoire des Sciences du Climat et de l'Environnement (LSCE-IPSL), CEA-CNRS-UVSQ, Université Paris-Saclay, F-91191 Gif-sur-Yvette, France
[6] Finnish Meteorological Institute (FMI), Helsinki, Finland
[7] Swiss Federal Laboratories for Materials Science and Technology (Empa), Dübendorf, Switzerland
[8] Energy research Centre of the Netherlands (ECN), Petten, Netherlands
[9] National Physical Laboratory, Teddington, Middlesex, TW11 0LW, UK
[10] School of GeoSciences, The University of Edinburgh, Edinburgh, EH9 3FF, UK
[11] Institut für Umweltphysik, Heidelberg University, Germany
[12] Center for Isotope Research (CIO), University of Groningen, Netherlands
[13] Cooperative Institute for Research in Environmental Sciences (CIRES), University of Colorado, Boulder, CO, USA
[14] Hungarian Meteorological Service, Budapest, Hungary
[15] Research Centre for Astronomy and Earth Sciences, Geodetic and Geophysical Institute, Sopron, Hungary
[16] Norwegian Institute for Air Research (NILU), Norway
[17] Umweltbundesamt, Messstelle Schauinsland, Kirchzarten, Germany
[18] AGH University of Science and Technology, Krakow, Poland
[19] Atmospheric Chemistry Research Group, University of Bristol, Bristol, UK
[20] Voeikov Main Geophysical Observatory, St. Petersburg, Russia
[21] NOAA Earth System Research Laboratory, Global Monitoring Division, Boulder, CO, USA

*Correspondence to*: Peter Bergamaschi (peter.bergamaschi@ec.europa.eu)

**Abstract.** We present inverse modelling ('top-down') estimates of European methane ($CH_4$) emissions for 2006-2012 based on a new quality-controlled and harmonized in-situ data set from 18 European atmospheric monitoring stations. We applied an ensemble of seven inverse models and performed four inversion experiments, investigating the impact of different sets of stations and the use of 'a priori' information on emissions.

The inverse models infer total $CH_4$ emissions of 26.8 (20.2-29.7) Tg $CH_4$ $yr^{-1}$ (mean, 10th and 90th percentiles from all inversions) for the EU-28 for 2006-2012 from the four inversion experiments. For comparison, total anthropogenic $CH_4$ emissions reported to UNFCCC ('bottom-up', based on statistical data and emissions factors) amount to only 21.3 Tg $CH_4$ $yr^{-1}$ (2006) to 18.8 Tg $CH_4$ $yr^{-1}$ (2012). A potential explanation for the higher range of 'top-down' estimates compared to 'bottom-up' inventories could be the contribution from natural sources, such as peatlands, wetlands, and wet soils. Based on seven

different wetland inventories from the "Wetland and Wetland $CH_4$ Inter-comparison of Models Project" (WETCHIMP) total wetland emissions of 4.3 (2.3-8.2) $CH_4$ $yr^{-1}$ from EU-28 are estimated. The hypothesis of significant natural emissions is supported by the finding that several inverse models yield significant seasonal cycles of derived $CH_4$ emissions with maxima in summer, while anthropogenic $CH_4$ emissions are assumed to have much lower seasonal variability. Taking into account the wetland emissions from the WETCHIMP ensemble, the top-down estimates are broadly consistent with the sum of

anthropogenic and natural bottom-up inventories. However, the contribution of natural sources remains rather uncertain, especially their regional distribution.

Furthermore, we investigate potential biases in the inverse models by comparison with regular aircraft profiles at four European sites and with vertical profiles obtained during the "Infrastructure for Measurement of the European Carbon Cycle (IMECC)" aircraft campaign. We present a novel approach to estimate the biases in the derived emissions, based on the comparison of

20 simulated and measured enhancements of $CH_4$ compared to the background, integrated over the entire boundary layer and over the lower troposphere. The estimated average regional biases range between -40% and 20% at the aircraft profile sites in France, Hungary and Poland.

## 1 Introduction

Atmospheric methane ($CH_4$) is the second most important long-lived anthropogenic greenhouse gas (GHG), after carbon

dioxide ($CO_2$), and contributed ~17% to the direct anthropogenic radiative forcing of all long-lived GHGs in 2016, relative to 1750 (NOAA Annual Greenhouse Gas Index (AGGI) [*Butler and Montzka*, 2017]). The globally averaged $CH_4$ mole fraction reached a new high of $1842.8 \pm 0.5$ ppb in 2016 (global average from marine surface sites [*Dlugokencky*, 2017]), more than 2.5 times the pre-industrial level [*WMO*, 2016b]. The increase in atmospheric $CH_4$ has been monitored by direct atmospheric measurements since the late 1970s [*Blake and Rowland*, 1988; *Cunnold et al.*, 2002; *Dlugokencky et al.*, 1994; *Dlugokencky*

*et al.*, 2011]. Atmospheric growth rates were large in the 1980s, decreased in the 1990s and were close to zero during 1999-2006. Since 2007, atmospheric $CH_4$ increased again significantly [*Dlugokencky et al.*, 2009; *Nisbet et al.*, 2014; *Rigby et al.*, 2008], at an average growth rate of $5.7 \pm 1.1$ ppb $yr^{-1}$ during 2007-2013, and at a further increased rate of $10.1 \pm 2.3$ ppb $yr^{-1}$ during 2014-2016 [*Dlugokencky*, 2017].

While the global net balance (global sources minus global sinks) of $CH_4$ is well defined by the atmospheric measurements of

35 in-situ $CH_4$ mole fractions at global background stations, the attribution of the observed spatial and temporal variability to specific sources and regions remains very challenging [*Houweling et al.*, 2017; *Kirschke et al.*, 2013; *Saunois et al.*, 2016]. Global inverse models are widely used to estimate emissions of $CH_4$ at global/continental scale, using mainly high-accuracy surface measurements at remote stations (e.g. [*Bergamaschi et al.*, 2013; *Bousquet et al.*, 2006; *Mikaloff Fletcher et al.*, 2004a; b; *Saunois et al.*, 2016]). In addition, satellite retrievals of GHGs have also been used in a number of studies. In particular,

near-IR retrievals from SCIAMACHY and GOSAT providing column average mole fractions ($XCH_4$) have been demonstrated

to provide additional information on the emissions at regional scales [*Alexe et al.*, 2015; *Bergamaschi et al.*, 2009; *Wecht et al.*, 2014]. However, current satellite retrievals may still have biases and their use in atmospheric models is at present limited by the shortcomings of models in realistically simulating the stratosphere, especially at higher latitudes [*Alexe et al.*, 2015; *Locatelli et al.*, 2015]. Furthermore, integration over the entire column implies that the signal from the $CH_4$ variability in the planetary boundary layer (which is directly related to the regional emissions) is reduced in the retrieved $XCH_4$.

In contrast, in-situ measurements at regional surface monitoring stations can directly monitor the atmospheric mole fractions within the boundary layer, providing strong constraints on regional emissions. Such regional monitoring stations have been set up in the last years especially in the United States [*Andrews et al.*, 2014] and Europe (e.g., [*Levin et al.*, 1999; *Lopez et al.*, 2015; *Popa et al.*, 2010; *Schmidt et al.*, 2014; *Vermeulen et al.*, 2011]). The measurements from these stations were used in a number of inverse modelling studies to estimate emissions at regional and national scales [*Bergamaschi et al.*, 2010; *Bergamaschi et al.*, 2015; *Ganesan et al.*, 2015; *Henne et al.*, 2016; *Kort et al.*, 2008; *Manning et al.*, 2011; *Miller et al.*, 2013]. A specific objective of these studies is the verification of 'bottom-up' emission inventories reported under the United Nations Framework Convention on Climate Change (UNFCCC), which are based on statistical activity data and measured or estimated emission factors [*IPCC*, 2006]. For many $CH_4$ source sectors (e.g., fossil fuels, waste, agriculture), emission factors exhibit large spatial, temporal and site-to-site variability (e.g., *Brandt et al.* [2014]), which inherently limits the capability of bottom-up approaches to provide accurate total emissions. Particular challenges are the representation of 'high-emitters' or 'super-emitters' in bottom-up inventories [*Zavala-Araiza et al.*, 2015], but also of minor source categories (e.g., abandoned coal mines or landfill sites), which, if not properly accounted for, may result in incorrect inventories. Independent verification using atmospheric measurements and inverse modelling is therefore considered essential to ensure the environmental integrity of reported emissions [*Levin et al.*, 2011; *National Academy of Science*, 2010; *Nisbet and Weiss*, 2010; *Weiss and Prinn*, 2011] and has been suggested to be used for the envisaged 'transparency framework' under the Paris agreement [*WMO*, 2016a].

Inverse modelling ('top-down') is a mass-balance approach, providing information from the integrated emissions from all sources. However, the quality of the derived emissions critically depends on the quality and density of measurements, and the quality of the atmospheric models used. In particular, when aiming at verification of bottom-up inventories, thorough validation of inverse models and realistic uncertainty estimates of the top-down emissions are essential.

*Bergamaschi et al.* [2015] showed that the range of the derived total $CH_4$ emissions from north-western and eastern Europe using four different inverse modelling systems, was considerably larger than the uncertainty estimates of the individual models. While the latter typically use Bayes' theory to calculate the reduction of assumed 'a priori' emission uncertainties by assimilating measurements (propagating estimated observation and model errors to the estimated emissions), an ensemble of inverse models may provide more realistic overall uncertainty estimates, since estimates of model errors are often based on strongly simplified assumptions and do not represent the total uncertainty. Furthermore, validation of the inverse models against independent observations not used in the inversion is important to assess the quality of the inversions.

Here, we present a new analysis, estimating European $CH_4$ emissions over the time period 2006-2012 using seven different inverse models. We apply a new, quality-controlled and harmonized data set of in-situ measurements from 18 European atmospheric monitoring stations generated within the European FP7 project InGOS ("Integrated non-$CO_2$ Greenhouse gas Observing System"). The InGOS data set is complemented by measurements from additional European and global discrete air sampling sites. Compared to the previous paper by *Bergamaschi et al.* [2015], which analysed 2006-2007, this study extends the target period (2006-2012), takes advantage of the larger and more stringently quality-controlled observational data set, and includes additional inverse models. Furthermore, we present a more comprehensive validation of model results using an extended set of aircraft observations, aiming at a more quantitative assessment of the overall errors. Finally we examine in more detail the potential contribution of natural emissions (such as peatlands, wetlands, or wet soils) using seven different

wetland inventories from the "Wetland and Wetland CH₄ Inter-comparison of Models Project" (WETCHIMP) [*Melton et al.*, 2013; *Wania et al.*, 2013].

## 2 Atmospheric measurements

The European monitoring stations used in this study are compiled in Table 1 and their locations are shown in Figure 1. The core data set is from 18 stations with in-situ $CH_4$ measurements. These measurements have been rigorously quality-controlled within the InGOS project. The quality control includes regular measurements of so-called target gases that monitor instrument performance and long-term stability [*Hammer et al.*, 2013; *Lopez et al.*, 2015; *Schmidt et al.*, 2014; *WMO*, 1993]. The instrument precision has been evaluated as 24 h moving 1σ standard deviation of bracketing working standards (denoted as "working standard repeatability"). A suite of other quality measures and error contributions, uncertainty in non-linearity corrections, potentially causing systematic biases between stations, have been investigated [*Vermeulen*, 2016], however, they have not been used in the inversions. The in-situ measurements are reported as hourly average dry air mole fractions (in units of nmol mol$^{-1}$, abbreviated as ppb), including the standard deviation of all individual measurements within one hour.

At most stations, the measurements have been performed using gas chromatography (GC) systems equipped with flame ionization detectors (FID). At the station Pallas (PAL), a GC-FID was applied until January 2009, and then replaced by a cavity ring-down spectrometer (CRDS). CRDS measurements (which are superior in precision compared to GC-FID) also started at other measurement sites, but here we used the GC measurements wherever available for the sake of time-series consistency while CRDS measurements were included for quality control and error assessment.

The InGOS measurements are calibrated against the NOAA-2004 standard scale (which is equivalent to the World Meteorological Organization Global Atmosphere Watch WMO-CH4-X2004 $CH_4$ mole fraction scale) [*Dlugokencky et al.*, 2005], except the InGOS measurements at Mace Head (MHD), for which the Tohoku University (TU) $CH_4$ standard scale has been used [*Aoki et al.*, 1992; *Prinn et al.*, 2000]. The two calibration scales are in close agreement. Based on parallel measurements by NOAA and Advanced Global Atmospheric Gases Experiment (AGAGE) at five globally distributed stations over more than 20 years an average difference of 0.3 ± 1.2 ppb between the two scales has been found. This difference is considered as not significant, and therefore no scale correction has been applied. In this study, we use the InGOS "release 2014" data set.

Six InGOS stations are equipped with tall towers, with uppermost sampling heights of 97-300 m above the surface, eight sites are surfaces stations (at low altitudes) with sampling heights of 6-60 m, and four sites are mountain stations (at altitudes between 1205 m and 3575 m asl).

The in-situ measurements at the InGOS stations are complemented by discrete air samples from the NOAA Earth System Research Laboratory (ESRL) global cooperative air sampling network at 11 European sites (and additional global NOAA sites used for the global inverse models) [*Dlugokencky et al.*, 1994; *Dlugokencky et al.*, 2009] and at five sites from the French RAMCES (Réseau Atmosphérique de Mesure des Composés à Effet de Serre) network [*Schmidt et al.*, 2006]. The discrete air measurements are taken from samples which are usually collected weekly.

For validation of the inverse models, we use $CH_4$ measurements of discrete air samples from four European aircraft profile sites at Griffin, Scotland (GRI), Orléans, France (ORL), Hegyhátsál, Hungary (HNG) and Bialystok, Poland (BIK) (see Figure 1). The analyses of the samples from GRI, ORL and HNG were performed at the Laboratoire des Sciences du Climat et de l' Environnement (LSCE) with the same GC used for RAMCES sites, those from BIK at the Max Planck Institute for Biogeochemistry (MPI).

Furthermore, we use airborne in-situ measurements from a campaign over Europe, which was performed in September / October 2009 as part of the "Infrastructure for Measurement of the European Carbon Cycle" (IMECC) project [*Geibel et al.*,

2012]. All measurements of the discrete air samples (from the NOAA and RAMCES surfaces sites and LSCE and MPI aircraft profile sites) and from the IMECC aircraft campaign are calibrated against the WMO-CH4-X2004 scale.

## 3 Modelling

### 3.1 Inversions

Four inversions were performed, investigating the impact of different sets of stations and the use of 'a priori' information on emissions (see Table 2). Inversion S1 covers 2006-2012 using a base set of observations (including only stations with maximum data gaps of 1 year), while inversions S2, S3, and S4 were performed for the years 2010-2012 and include additional stations, for which not all data are available before 2010. In S1, S2, and S3 the InGOS data set is used along with the discrete air samples from NOAA and RAMCES surfaces sites, while in S4 only the InGOS data are used. The exact sets of stations

applied in the different inversion experiments are indicated in Table 1. Inversion S1, S2, and S4 use 'a priori' information of $CH_4$ emissions from gridded inventories. For the anthropogenic $CH_4$ emissions, the "EDGARv4.2FT-InGOS" inventory is used, which integrates information on major point sources from the European Pollutant Release and Transfer Register (E-PRTR) into the EDGARv4.2FastTrack $CH_4$ inventory (http://edgar.jrc.ec.europa.eu/overview.php?v=ingos) [*Janssens-Maenhout et al.*, 2014]. Since EDGARv4.2FT-InGOS covers only the period 2000-2010, the inventory of 2010 has been

applied as 'a priori' also for 2011 and 2012. For the natural $CH_4$ emissions from wetlands, most models used the wetland inventory of J. Kaplan [*Bergamaschi et al.*, 2007] as 'a priori', except TM5-CTE, which applied LPX- Bern v1.0 [*Spahni et al.*, 2013] instead. Inversion S3 was performed without using detailed bottom-up inventories as 'a priori', in order to analyse the constraints of observed atmospheric $CH_4$ on emissions independent of 'a priori' information (using a homogeneous distribution of emissions over land and over the ocean, respectively, as starting point for the inversions in a similar manner as

in *Bergamaschi et al.* [2015]; for further details see section 1 of supplementary material).

### 3.2 Atmospheric models

The atmospheric models used in this study are listed in Table 3. The models include global Eulerian models with zoom over Europe (TM5-4DVAR, TM5-CTE, LMDZ), regional Eulerian models (CHIMERE) and Lagrangian dispersion models (STILT, NAME, COMET). The horizontal resolutions over Europe are ~1.0-1.2° (longitude) × ~0.8-1.0° (latitude) for the

global models (zoom), and ~0.17-0.56° (longitude) × ~0.17-0.5° (longitude) for the regional models. The regional models use boundary conditions (background $CH_4$ mole fractions) from inversions of the global models (STILT from TM3, COMET from TM5-4DVAR, CHIMERE from LMDZ, or estimate the boundary conditions in the inversions (NAME), using baseline observations at Mace Head as 'a priori' estimates. In case of NAME and CHIMERE, the boundary conditions are further optimized in the inversion.

All models used the same observational data set described in section 2 (except the stations ZEP and ICE, that are outside the domain of some regional models and except the mountain stations JFJ, PDM and KAS, which were not used in the NAME inversions). For the stations with in-situ measurements in the boundary layer, most models assimilated only measurements in the early afternoon (between 12:00 and 15:00 LT), and for mountain stations only night-time measurements (between 00:00 and 03:00 LT) [*Bergamaschi et al.*, 2015]. However, NAME and COMET used observations at all times. The individual

inverse models are described in more detail in the supplementary material (section 1).

## 4 Results and discussion

### 4.1 European CH$_4$ emissions

Figure 2 shows the maps of the European CH$_4$ emissions (average 2010-2012) derived from the seven inverse models for inversion S4. The corresponding maps for inversions S1-S3 (available from five models) are shown in the supplementary material (Figures 1S-3S). In S1, S2, and S4, which are guided by the 'a priori' information from the emission inventories, the 'a posteriori' spatial distributions are usually close to the prior patterns on smaller scales (determined by the chosen spatial correlation scale lengths). The NAME inversion groups together grid cells for which the observational constraints are weak, i.e., it averages over increasingly larger areas at larger distances from the observations. Consequently, in the NAME inversion the 'fine structure' of the 'a priori' inventories disappears in areas which are not well constrained (e.g., Spain).

Comparing inversions S1, S2, and S4 shows overall very similar spatial patterns for all inverse models, indicating only moderate differences in the observational constraints of the three different sets of stations. In particular, addition of NOAA and RAMCES discrete air samples (inversion S2 vs. S4) results in only minor differences in the derived emissions. When the larger set of InGOS stations (S2 vs. S1) is used, most models yield higher CH$_4$ emissions from Northern Italy. This is most likely mainly due to the observations from Ispra (IPR), at the north-western edge of the Po valley, while this area is not well constrained in S1.

The information content of the observations is further examined in inversion S3, which does not use detailed emission inventories (Figure 3S), similar to a previous sensitivity experiment in *Bergamaschi et al.* [2015]. Especially TM5-4DVAR and TM3-STILT yield similar spatial distributions with elevated CH$_4$ emissions from the BENELUX area and northwestern Germany, from the coastal area of northwestern France, Ireland, UK, and the Po valley. Most of these patterns are visible also in inversion S3 of NAME, however with more variability on smaller scales (while TM5-4DVAR and TM3-STILT show much smoother distributions). These regional hotspots are broadly consistent with the bottom-up inventories, which illustrates the principal capability of inverse modelling to derive emissions that are independent of detailed 'a priori' inventories in the vicinity of observations. LMDZ and TM5-CTE also show elevated emissions over western and central Europe, but in contrast to the other three inverse models no regional hotspots. For TM5-CTE this is related to the applied inversion technique (adjusting emissions uniformly over large predefined regions), which effectively limits the number of degrees of freedom, and does not allow retrieval of regional hotspots, if such patterns are not 'a priori' present within the predefined regions. For LMDZ, the lack of regional hotspots is probably related to the specific settings for this scenario, with a spatial correlation scale length of 500 km, significantly larger than in TM5-4DVAR (50 km) and TM3-STILT (60 km).

Figure 3a displays the annual total European CH$_4$ emissions derived by the models for 2006-2012 in inversion S1, and for 2010-2012 in S2-S4. The figure shows the total emissions from all EU-28 countries, and separately from northern Europe (Norway, Sweden, Finland, Baltic countries, and Denmark), western Europe (UK, Ireland, Netherlands, Belgium, Luxembourg, France, Germany, Switzerland, and Austria), eastern Europe (Poland, Czech Republic, Slovakia, and Hungary), and southern Europe (Portugal, Spain, Italy, Slovenia, Croatia, Greece, Romania, and Bulgaria). The non-EU-28 countries Norway and Switzerland are included here in 'northern Europe' and 'western Europe', respectively, but not in EU-28. Six of the seven models yield considerably higher total CH$_4$ emissions from the EU-28 compared to the anthropogenic CH$_4$ emissions reported to UNFCCC (submission 2016), while NAME is very close to the UNFCCC emissions. This behaviour is apparent also for the European subregions western, eastern and southern Europe, while for northern Europe (where natural CH$_4$ emissions play a large role) also NAME yields higher total CH$_4$ emissions compared to UNFCCC (except for S3 in 2011 and 2012).

Figure 3a also shows the results from the previous study of *Bergamaschi et al.* [2015], which used four inverse models (previous versions of those applied in this study) and a set of 10 European stations with continuous measurements

(complemented by discrete air samples) to estimate $CH_4$ emissions in 2006-2007. For TM5-4DVAR, TM3-STILT, and LMDZ the results are relatively similar (within ~10% for EU-28) to this study, while the $CH_4$ emissions from NAME were ~20% lower (EU-28). Despite the significantly larger number of European monitoring stations in the present study, however, we emphasize that the available stations do not very well cover the whole EU-28 area. Consequently, the emissions especially
from Southern Europe remain poorly constrained.

For comparison of total emissions derived by the inverse models and anthropogenic emissions from emission inventories it is essential to account for natural emissions, especially from wetlands, peatlands and wet soils. As an estimate of these emissions and their uncertainties, we use an ensemble of seven wetland inventories from the "Wetland and Wetland $CH_4$ Inter-comparison of Models Project" (WETCHIMP) [*Melton et al.*, 2013; *Wania et al.*, 2013] (the spatial distribution of European $CH_4$ emissions
from the different individual WETCHIMP inventories is shown in Figure 4S). Figure 3a shows the mean, median, minimum and maximum $CH_4$ emissions from this ensemble for EU-28 and the different European subregions. These quantities are evaluated after integrating over the corresponding areas, using the multi-annual mean (1993-2004) of the WETCHIMP inventories. For northern Europe, in particular, the estimated wetland emissions are high (2.5 (1.7-4.3) Tg $CH_4$ yr$^{-1}$ (mean, minimum, maximum)) and exceed the anthropogenic $CH_4$ emissions (UNFCCC: 1.3 Tg $CH_4$ yr$^{-1}$; mean 2006-2012).
Substantial wetland emissions are also estimated for western Europe (1.6 (0.4-3.1) Tg $CH_4$ yr$^{-1}$), but wetland emissions are also non-negligible for eastern Europe (0.3 (0.03-0.9) Tg $CH_4$ yr$^{-1}$) and southern Europe (0.6 (0.01-1.1) Tg $CH_4$ yr$^{-1}$), especially when considering the upper range of these estimates. For EU-28, wetland emissions of 4.3 (2.3-8.2) Tg $CH_4$ yr$^{-1}$ are estimated, corresponding to 22% (11%-41%) of reported anthropogenic $CH_4$ emissions.

Taking into account the estimates of the WETCHIMP ensemble brings the results of the six inverse models that derive high
emissions into the upper uncertainty range of the sum of anthropogenic emissions (reported to UNFCCC) and wetland emissions, while the emissions derived by NAME fall in the lower range (Figure 3b). This analysis suggests broad consistency between bottom-up and top-down emission estimates, albeit with a clear tendency (6 of 7 models) towards the upper range of the bottom-up inventories for the total $CH_4$ emissions from EU-28. This behaviour is apparent also for western and southern Europe, while for eastern Europe several models are close to or above the upper uncertainty bound (NAME is very close to
the mean), and for northern Europe several models are rather in the lower range (or below the lower uncertainty bound) of the combined UNFCCC and WETCHIMP inventory.

Critical to the assessment of consistency between the different approaches, is the analysis of their uncertainties. Inverse models usually propagate estimated observation and model errors to the estimated emissions, however in particular the model errors are generally based on simplified assumptions. Furthermore, the error estimates of the inverse models take usually only random
errors into account, and are based on the assumption that observation and model errors are unbiased. Estimated $2\sigma$ uncertainties for EU-28 top-down emissions range between ~7% and ~33% (except for inversion S3 of NAME, for which uncertainties are larger than 50%). For the subregions 'northern Europe' and 'southern Europe', which are poorly constrained by measurements, the model estimates of the relative uncertainties are significantly larger, ranging between ~20% and more than ~100%.

The ($2\sigma$) uncertainties of the UNFCCC inventories shown in Figure 3a are based on the uncertainties of major $CH_4$ source
categories reported by the countries in their national inventory reports. To calculate the uncertainties of total emissions per country (or group or countries), the reported uncertainties per category were aggregated as described in *Bergamaschi et al.* [2015]. We note, however, that uncertainties reported for the same category by different countries exhibit large differences (e.g., for coal between 9 and 300%, for oil and natural gas between 5 and 460%, for enteric fermentation between 7 and 50%, for manure management between 5 and 100%, and for solid waste disposal between 22 and 126%), with the lower uncertainty
estimates appearing unrealistically low. Furthermore, the estimates of the total uncertainties consider only the major categories

(EU-28: 93% of reported emissions) and do not take into account potential additional emissions (and their uncertainties) that are not covered by the inventories.

Figure 3a includes also the anthropogenic $CH_4$ emissions from EDGARv4.2FT-InGOS (for 2006-2010), which are at the upper uncertainty bound of the UNFCCC inventories for EU-28. The difference between UNFCCC and EDGAR is mainly due to significant differences in $CH_4$ emissions from fossil fuels (coal, oil, and natural gas), which, however, might be overestimated in some cases in EDGAR [*Bergamaschi et al.*, 2015].

For wetlands, very large differences between the different inventories of the WETCHIMP ensemble are apparent regarding the spatial emission distribution (see Figure 4S) and the magnitude of the emissions, illustrating the very high uncertainties in the current estimates. Comparing the different wetland inventories, a striking pattern is visible for LPJ-WHyMe, with very high $CH_4$ emissions for the British Isles. The climate of this region has mild winters that allow simulated wetland $CH_4$ emissions to continue year-round, yielding high annual emissions intensity for LPJ-WHyMe [*Melton et al.*, 2013].

In the previous analysis of *Bergamaschi et al.* [2015] the contribution from natural sources in western and eastern Europe was considered to be very small, based on the wetland inventory of J. Kaplan [*Bergamaschi et al.*, 2007]. However, that inventory is close to the lower estimates of the WETCHIMP ensemble. Unfortunately, direct comparisons of $CH_4$ emissions simulated by the different wetland inventories with local or regional $CH_4$ flux measurements in European wetland areas are lacking. Therefore, no conclusions can be drawn as to which of the inventories is most realistic.

To further investigate the contribution of wetland emissions we analyse the seasonal variations. Figure 4 illustrates that four inverse models (TM5-4DVAR, TM5-CTE, TM3-STILT, and LMDZ) calculate pronounced seasonal variations in total emissions. For EU-28 the derived seasonality is largely consistent with the seasonality of the wetland emissions from the WETCHIMP ensemble (both regarding the amplitude, and the phase with maxima in summer). For northern Europe the seasonal variations derived by the four inverse models are somewhat smaller compared to the mean of the WETCHIMP ensemble, while for western and eastern Europe they are somewhat larger, but still broadly within the minimum-maximum range of the WETCHIMP inventories. For southern Europe, the seasonality of the four inverse models is more irregular, and the maximum emissions for the wetland ensemble show a clear peak in winter, which however is not apparent in the mean or median of the ensemble. This is probably due to the important role of precipitation for the wetland emissions in southern Europe, while for temperate and boreal regions the seasonal variation of wetland emissions is mainly driven by temperature (e.g., [*Christensen et al.*, 2003; *Hodson et al.*, 2011]). In contrast to the discussed four models, NAME derives much smaller seasonal variations, and for western Europe, eastern Europe, and EU-28 with opposite phase (small maximum in winter). Only for northern Europe, also NAME estimates maximum emissions in summer, however the amplitude is much smaller compared to the other models and the WETCHIMP wetland inventories. One reason contributing to the smaller amplitude is that NAME provides only 3-monthly emissions (compared to monthly resolution of the other four inverse models), but the lower temporal resolution of NAME clearly explains only a smaller part of the different seasonal cycles. Figure 5S shows that also in inversion S3 (which is not using any detailed a priori inventory nor any a priori seasonal cycle) significant seasonal cycles of $CH_4$ emissions are derived by TM5-4DVAR, TM3-STILT, LMDZ, and TM5-CTE, which demonstrates that the derived seasonal cycles are mainly driven by the observations, and not by the a priori.

Apart from the different behaviour of NAME, the finding that four inverse models derive seasonal cycles that are broadly consistent with the seasonal cycles calculated by the WETCHIMP ensemble supports a significant contribution of wetlands to the total $CH_4$ emissions. Commonly, anthropogenic $CH_4$ emissions are assumed to have no significant seasonal variations, except $CH_4$ emissions from rice and biomass burning (which however play only a minor role in Europe). Unfortunately, only very limited information is available about potential seasonal variations of anthropogenic $CH_4$ sources (other than rice and biomass burning). *Ulyatt et al.* [2010] reported significant seasonal variations of $CH_4$ emissions from dairy cows, mainly

related to the lactation periods of cows. *VanderZaag et al.* [2014], estimating total $CH_4$ emissions from two dairy farms, found higher $CH_4$ emissions in fall compared to spring, mainly due to varying $CH_4$ emissions from manure management. Beside agricultural $CH_4$ sources, $CH_4$ from landfills [*Spokas et al.*, 2011] and waste water may also exhibit seasonal variations, while only small seasonal variations were found for natural gas distribution systems [*McKain et al.*, 2015; *Wennberg et al.*, 2012; *Wong et al.*, 2016] (and further references therein). Quantitative estimates of potential seasonal variations of anthropogenic sources cannot be made due to the limited number of studies, but the relative variability of the total anthropogenic sources is expected to be much smaller compared to wetlands.

Model simulations and bottom-up inventories for individual countries (or group of countries) are shown in the supplementary material (Figure 6S), illustrating further that wetland emissions are important, particularly in northern European countries, but may also contribute significantly for many other countries.

Finally, we analyse the trends in $CH_4$ emissions (Figure 7S). Anthropogenic $CH_4$ emissions reported to UNFCCC for EU-28 decreased by $-0.44 \pm 0.02$ Tg $CH_4$ yr$^{-2}$ during 2006-2012. Also all 5 inversions which are available for this period (inversion S1) derive negative $CH_4$ emission trends ranging between $-0.19$ and $-0.58$ Tg $CH_4$ yr$^{-2}$. The uncertainties given for the trends of the individual inversions (and the reported $CH_4$ emissions), however, include only the uncertainty of the linear regression (i.e. reflecting the scatter of the annual values around the linear trend), but do not take into account the uncertainties of the annual mean values and the error correlations between different years. In particular the latter remain very difficult to estimate, which currently limits clear conclusions about the significance of the trends.

**4.2 Evaluation of inverse models**

First we evaluate the performance of model simulations at the atmospheric monitoring stations. Figure 8S shows the correlation coefficients, bias, root mean square (RMS) difference, and the ratio between modelled and observed standard deviation for inversion S4, including stations that were assimilated and stations that were used for validation only. For the evaluation of the statistics for the in-situ measurements, we use only early afternoon data (between 12:00 and 15:00 LT). Averaging over all stations, the correlation coefficients are between 0.65 and 0.79 for 6 models, and 0.5 for COMET. The ranking of models in terms of correlation coefficients is closely reflected in the achieved average RMS values, ranging between 33 and 70 ppb (with models with higher correlation coefficients typically achieving lower average RMS). At several tall towers a clear tendency of decreasing RMS with increasing sampling height is visible, demonstrating the benefit of higher sampling heights, which allow more representative measurements that are less affected by local sources and that can be better reproduced by the models.

While the evaluation of the model simulations at the monitoring stations provides a measure of the quality of the inversions and the atmospheric transport models applied (e.g., with the correlation coefficients describing how much of the observed variability can be explained by the models), the analysis of the station statistics cannot quantify how realistic the derived emissions are, but gives only some qualitative indications about potential biases of the emissions. The inverse models optimize model emissions to achieve an optimal agreement between simulated and observed atmospheric $CH_4$ mole fractions (taking into account the a priori constraints). This implies that potential biases of the model (or the observations) may be compensated in the inversions by introducing biases in the derived emissions. In particular, vertical mixing of the models is very critical in this context. For example, too strong vertical mixing of the transport models may be compensated in the inversion by enhancing the model emissions (i.e. deriving model emissions that are higher than real emissions) such that a good agreement between simulated and observed mole fractions at the surface can still be achieved. An important diagnostic to identify such potential systematic errors is the analysis of vertical profiles (including the boundary layer and the free troposphere). For this purpose we compare our model simulations with regular aircraft profiles at four European sites (Figure 5). At Griffin (GRI), observed and simulated mole fractions show only small vertical gradients, while at Orléans (ORL), Hegyhátsál (HNG), and Bialystok

(BIK) large vertical gradients are visible, with increasing values towards the surface. The figure also includes the background mole fractions in the absence of model emissions over Europe calculated by TM5-4DVAR (based on the scheme of *Rödenbeck et al.* [2009]). At GRI, the measurements are in general very close to the background mole fractions, illustrating that the impact of European emission is rather limited at this site. In contrast, pronounced enhancements in measured and simulated $CH_4$

compared to the background are apparent at the other three sites, especially in the lower ~2 km due to regional emissions. These enhancements show some seasonal variation, with largest vertical extension during summer (~2 km), while they are confined to the lower ~1 km during winter, due to the seasonal variations in the average boundary layer height [*Koffi et al.*, 2016]. Please note that the differences of the background mole fractions which are visible in Figure 5 between some sites, are partly due to the different temporal sampling at the different sites (compare Figure 6).

To analyse potential model biases more quantitatively, we evaluate in the following the enhancement of observations and model simulations compared to background $CH_4$ values (1) integrated over the entire boundary layer, and (2) integrated over the lower troposphere up to ~3-4 km. The rationale behind this approach is that emissions initially mainly accumulate within the boundary layer. Therefore, potential biases in model emissions should be reflected in differences between the observed and modelled integrated enhancement within the boundary layer. For the overall budget, however, mixing between boundary

layer and free troposphere plays an important role. Thus, the enhancement integrated over the entire lower troposphere provides additional diagnostics for potential model biases.

The integration of the enhancements is shown for the individual profiles at ORL, HNG and BIK in the supplementary material (Figures 9S, 10S, 11S). In addition, we use also aircraft measurements from the IMECC campaign in September / October 2009 (Figure 12S). These include profile measurements at Orléans and Bialystok, but also at Karlsruhe, Jena, and Bremen,

hence extending the spatial coverage of the sites with regular profiles (ORL, HNG and BIK). To calculate the enhancements for the individual profiles, we apply the background mole fractions calculated for the TM5-4DVAR zoom domain as the common reference for the observations and the model simulations for all global models (i.e. TM5-4DVAR, TM5-CTE, and LMDZ). For STILT and NAME, the background $CH_4$ is calculated for the STILT and NAME domains, but the dependence of the background mole fractions (calculated by TM5-4DVAR) on the exact extension of the domain is generally rather small.

However, the $CH_4$ background mole fractions used in the inversions of the regional models (for NAME based on baseline observations at Mace Head and for TM3-STILT based on the TM3 model) shows significant differences compared to the TM5-4DVAR background, with typically ~10 ppb higher values at the three continental aircraft sites (ORL, HNG, BIK; see Figure 5). In order to investigate which background mole fractions are more realistic we compared the model simulations with the aircraft observations for events with very low simulated contribution ($\leq$ 3 ppb) from European $CH_4$ emissions (Figure 14S).

This analysis shows that TM5-4DVAR simulations are close to the observations (average bias between -1.1 and 3.5 ppb), which indicates that the TM5-4DVAR background is relatively realistic, while NAME and TM3-STILT are consistently higher at the continental aircraft sites with average biases of 12-13 ppb for NAME and 9-12 ppb for TM3-STILT. This supports the use of the background calculated with TM5-4DVAR as reference for the measurements. For the evaluation of the simulated $CH_4$ enhancements of the regional models, however, we use the actual background used in NAME and TM3-STILT.

For the integration over the boundary layer, we use the boundary layer height (BLH) diagnosed by TM5. A recent comparison of the TM5 BLH with observations from the NOAA Integrated Global Radiosonde Archive (IGRA) [*Koffi et al.*, 2016] showed that TM5 reproduces the daytime BLH relatively well (within ~10–20%), but larger deviations were found for the nocturnal BLH, especially during summer, when very low BLHs (< 100 m) are observed. Here, we use only profiles for which the (TM5 diagnosed) BLH is not lower than 500 m. The average enhancement of the measurements and model simulations in the

boundary layer compared to background is denoted as $\Delta c_{MOD, BL}$ and $\Delta c_{OBS, BL}$, respectively (further details about the evaluation of the enhancements are given in the supplementary material). Figure 6 shows the derived 'relative bias', defined as:

$$rb_{BL} = (\Delta c_{MOD,\,BL} - \Delta c_{OBS,\,BL}) / \Delta c_{OBS,\,BL}$$

for ORL, HNG, BIK for the entire target period 2006-2012 (inversion S1). The three global inverse models (i.e. TM5-4DVAR, TM5-CTE, and LMDZ) show in general only a small average relative bias ($rb_{BL}$ between -7% and 10%) at the three aircraft sites. In contrast, TM3-STILT and NAME have significant negative relative biases (TM3-STILT: $rb_{BL}$ between -13% and -24% for the three sites; NAME $rb_{BL}$ = -30% for ORL and HNG).

These negative biases are likely related to the positive bias in the background $CH_4$ used for NAME and TM3-STILT (see above), since the regional models invert the difference between the observations and the assumed background. In fact, also at most continental atmospheric monitoring stations, the background used for NAME and TM3-STILT is significantly higher (~10 ppb) compared to the TM5-4DVAR background (Figure 15S).

The 'relative bias' is also extracted separately for different seasons (right panel of Figure 6). There is no clear seasonal cycle in the relative bias apparent and the variability between the different seasons is generally small (data points at BIK for DJF are considered not significant as they are from one single profile only). From this analysis there is no evidence that the seasonal cycle of emissions derived by four inverse models (TM5-4DVAR, TM5-CTE, TM3-STILT, and LMDZ; see section 4.1) with clear maxima in summer could be due to a seasonal bias in the transport models. At the same time, however, NAME, which calculates much smaller seasonal variations of emissions, also shows no seasonal variations of the average bias at ORL and HNG. However, especially at HNG the total number of profiles is rather small (n=22), which limits the analysis of potential seasonal transport biases.

Figure 13S shows the relative bias of the $CH_4$ enhancements integrated over the lower troposphere, defined as:

$$rb_{COL} = (\Delta c_{MOD,\,COL} - \Delta c_{OBS,\,COL}) / \Delta c_{OBS,\,COL}.$$

The three global inverse models (i.e. TM5-4DVAR, TM5-CTE, and LMDZ) have a relative bias between of -4% and 20% at the three aircraft sites, indicating a small tendency to overestimate the European $CH_4$ emissions, while the regional models show a negative relative bias (TM3-STILT: between -9% and -20% for the three sites; NAME -31% for ORL and -40% HNG). Figure 7 presents an overview of the derived relative biases for the enhancement integrated over the boundary layer ($rb_{BL}$, top panel of figure) and in the lower troposphere ($rb_{COL}$, lower panel). The differences of the relative bias integrated over the lower troposphere compared to that integrated only over the boundary layer (e.g., $rb_{COL} > rb_{BL}$ for TM5-4DVAR and TM5-CTE at ORL and BIK) suggests that shortcomings of the models to simulate the exchange between the boundary layer and the free troposphere may contribute significantly to the bias in the derived emissions. An illustrative example of the shortcomings of the models to simulate the free troposphere are the IMECC profiles at Bialystok on 30 September 2009 (Figure 12S). The measurements show a considerable $CH_4$ enhancement (~25 ppb) at around 3.5 to 4 km, which is not reproduced by the models. This could indicate that cloud convective transport was missed by the models.

A general limitation of the analysis of the enhancements integrated over the lower troposphere, however, is that this analysis is more sensitive to potential errors in the simulated background mole fractions in the free troposphere compared to the boundary layer, because of the generally much lower enhancements in the free troposphere.

Finally, we analyse the correlation between the relative bias of the integrated $CH_4$ enhancements and the regional model emissions. Figure 16S shows the relationship between $rb_{BL}$ and the average model emissions around the aircraft site, integrating all model grid cells with a maximum distance of 400 km (hereafter referred to as integration radius) from the aircraft site. At all three sites clear correlations between $rb_{BL}$ and the regional model emissions are found, which confirms that $rb_{BL}$, derived from the aircraft profiles, can be used to diagnose biases in the regional model emissions.

The derived correlations depend on the chosen area, over which model emissions are integrated. For ORL and HNG, significant correlations were found for integration radii between 200 and 800 km, while for BIK different integration radii resulted in poorer correlations (now shown), probably related to significant differences in the spatial emission patterns derived by the

different models around this site. To further improve the analysis, the 'footprints' (i.e. sensitivities of atmospheric concentrations to surface emissions) of the individual aircraft profiles should be taken into account in the future. Furthermore, it would be useful, to calculate for all global models individually the background mole fractions using the scheme of *Rödenbeck et al.* [2009]. This would allow to derive the modelled $CH_4$ enhancements more accurately.

## 5 Conclusions

We have presented estimates of European $CH_4$ emissions for 2006-2012 using the new InGOS data set of in-situ measurements from 18 European monitoring stations (and additional discrete air sampling sites) and an ensemble of seven different inverse models. For the EU-28, total $CH_4$ emissions of 26.8 (20.2-29.7) Tg $CH_4$ yr$^{-1}$ are derived (mean, 10% percentile, and 90% percentile from all inversions), compared to total anthropogenic $CH_4$ emissions of 21.3 Tg $CH_4$ yr$^{-1}$ (2006) to 18.8 Tg $CH_4$ yr$^{-1}$ (2012) reported to UNFCCC. Our analysis highlights the potential significant contribution of natural emissions from wetlands (including peatlands and wet soils) to the total European emissions, with total wetland emissions of 4.3 (2.3-8.2) Tg $CH_4$ yr$^{-1}$ (EU-28) estimated from the WETCHIMP ensemble of seven different wetland inventories [*Melton et al.*, 2013; *Wania et al.*, 2013]. The hypothesis of a significant contribution from natural emissions is supported by the finding that four inverse models (TM5-4DVAR, TM5-CTE, TM3-STILT, LMDZ) derive significant seasonal variations of $CH_4$ emissions with maxima in summer. However, the NAME model calculates only a weak seasonal cycle, with small maximum (of EU-28 total $CH_4$ emissions) in winter. Furthermore, it needs to be emphasized that wetland inventories have large uncertainties and show large differences in the spatial distribution of $CH_4$ emissions.

Taking into account the estimates of the WETCHIMP ensemble, the bottom-up and top-down estimates of total EU-28 $CH_4$ emissions are broadly consistent within the estimated uncertainties. However, the results from six inverse models are in the upper uncertainty range of the sum of anthropogenic emissions (reported to UNFCCC) and wetland emissions, while the emissions derived by NAME are in the lower range. Furthermore, the comparison of bottom-up and top-down estimates shows some differences for the different European subregions. For northern Europe (including Norway) several models are rather in the lower range (or below the lower uncertainty bound) of the combined UNFCCC and WETCHIMP inventory, while for eastern Europe several models are close to the upper uncertainty bound or above (NAME is very close to the mean). Considering the estimated uncertainties of the inverse models, however, the uncertainty ranges of bottom-up and top-down estimates generally overlap for the different European subregions.

To estimate potential biases of the emissions derived by the inverse models, we analysed the enhancements of $CH_4$ mole fractions compared to the background, integrated over the entire boundary layer and over the lower troposphere, using regular aircraft profiles at four European sites and the IMECC aircraft campaign.

This analysis showed for the three global inverse models (TM5-4DVAR, TM5-CTE, and LMDZ) a relatively small average relative bias ($rb_{BL}$ between -7% and 10%, $rb_{COL}$ -4% and 20% for ORL, HNG and BIK). The regional models revealed a significant negative bias (TM3-STILT: $rb_{BL}$ between -13% and -24%, $rb_{COL}$ between -9% and -20% for ORL, HNG and BIK; NAME $rb_{BL}$ = -30%, $rb_{COL}$ between -31% and -40% at ORL and HNG). A potential cause for the negative relative bias of TM3-STILT and NAME is the significant positive bias of the background used in TM3-STILT (from global TM3 inversion) and NAME (based on measurements at baseline conditions at Mace Head).

The relative bias $rb_{BL}$ shows clear correlations with regional model emissions around the aircraft profile sites, which confirms that $rb_{BL}$ can be used to diagnose biases in the regional model emissions. The accuracy of the estimated relative biases, however, depends on the quality of the simulated background mole fractions. In particular the enhancements derived for the lower troposphere above the boundary layer (which are usually much smaller than the enhancements within the boundary

layer) are very sensitive to the background mole fractions. Therefore, potential model errors in the exchange between the boundary layer and the free troposphere (and their impact on the derived emissions) remain difficult to quantify.

Our study highlights the challenge to verify anthropogenic bottom-up emission inventories with the small uncertainties desirable for the international climate agreements. To reduce the uncertainties of the top-down estimates (1) the natural emissions need to be better quantified, (2) transport models need to be further improved, including their spatial resolution and in particular the simulation of vertical mixing, and (3) the network of atmospheric monitoring stations should be further extended, especially in southern Europe, which is currently clearly under-sampled. Furthermore, the uncertainty estimates of bottom-up inventories (including both the anthropogenic and natural emissions) and atmospheric inversions need to be further improved.

**Acknowledgements**

This work has been supported by the European Commission Seventh Framework Programme (FP7/2007–2013) project InGOS under grant agreement 284274. We thank Joe Melton for providing the WETCHIMP data set and for discussion about the wetland emission. We are grateful to Maarten Krol and Frank Dentener for helpful comments on the manuscript, and to Ernest Koffi for support of the further analyses of the boundary layer heights. ECMWF meteorological data have been preprocessed by Philippe Le Sager into the TM5 input format. We thank Arjo Segers for support of the TM5 modelling. Furthermore, we thank Johannes Burgstaller for the compilation of UNFCCC emission uncertainties. We are grateful to ECMWF for providing computing resources under the special project "Global and Regional Inverse Modeling of Atmospheric $CH_4$ and $N_2O$ (2012–2014)" and "Improve estimates of global and regional $CH_4$ and $N_2O$ emissions based on inverse modelling using in situ and satellite measurements (2015–2017) ". We thank François Marabelle and his team for computing support at LSCE.

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

**Tables**

**Table 1:** European monitoring stations used in this study. "s.h." is the sampling height (m) above ground, "ST" specifies the sampling type ("I": in-situ measurements; "D": discrete air sample measurements). The last four columns indicate the use of the corresponding station data set in the inversions S1-S4 (see section 3.1 and Table 2).

| ID | station name | data provider | lat | lon | alt | s. h. | ST | S1 | S2 | S3 | S4 |
|---|---|---|---|---|---|---|---|---|---|---|---|
| ZEP | Ny-Alesund | InGOS/NILU[1] | 78.91 | 11.88 | 474 | 15 | I | ● | ● | ● | ● |
| | | NOAA | 78.91 | 11.88 | 474 | 5 | D | ● | ● | ● | |
| SUM | Summit | NOAA | 72.60 | -38.42 | 3210 | 5 | D | ● | ● | ● | |
| PAL | Pallas | InGOS/FMI[2] | 67.97 | 24.12 | 565 | 7 | I | ● | ● | ● | ● |
| | | NOAA | 67.97 | 24.12 | 560 | 5 | D | ● | ● | ● | |
| ICE | Storhofdi, | NOAA | 63.40 | -20.29 | 118 | 9 | D | ● | ● | ● | |
| VKV | Voeikovo | InGOS/MGO[3] | 59.95 | 30.70 | 70 | 6 | I | | ● | ● | ● |
| TTA | Angus | InGOS/UoE[4] | 56.55 | -2.98 | 313 | 222 | I | ● | ● | ● | ● |
| BAL | Baltic Sea | NOAA | 55.35 | 17.22 | 3 | 25 | D | | | | |
| LUT | Lutjewad | InGOS/CIO[5] | 53.40 | 6.35 | 1 | 60 | I | ● | ● | ● | ● |
| MHD | Mace Head | InGOS/UoB[6] | 53.33 | -9.90 | 25 | 15 | I | ● | ● | ● | ● |
| | | NOAA | 53.33 | -9.90 | 5 | 21 | D | ● | ● | ● | |
| BIK1 | Bialystok | InGOS/MPI[7] | 53.23 | 23.03 | 183 | 5 | I | | | | |
| BIK2 | | | | | | 30 | I | | | | |
| BIK3 | | | | | | 90 | I | | | | |
| BIK4 | | | | | | 180 | I | | | | |
| BIK5 | | | | | | 300 | I | ● | ● | ● | ● |
| CBW1 | Cabauw | InGOS/ECN[8] | 51.97 | 4.93 | -1 | 20 | I | | | | |
| CBW2 | | | | | | 60 | I | | | | |
| CBW3 | | | | | | 120 | I | | | | |
| CBW4 | | | | | | 200 | I | ● | ● | ● | ● |
| OXK1 | Ochsenkopf | InGOS/MPI[7] | 50.03 | 11.82 | 1022 | 23 | I | | | | |
| OXK2 | | | | | | 90 | I | | | | |
| OXK3 | | | | | | 163 | I | ● | ● | ● | ● |
| OXK | | NOAA | 50.03 | 11.82 | 1022 | 163 | D | | | | |
| HEI | Heidelberg | InGOS/IUP[9] | 49.42 | 8.67 | 116 | 30 | I | ● | ● | ● | ● |
| KAS | Kasprowy Wierch | InGOS/AGH[10] | 49.23 | 19.98 | 1987 | 2 | I | | ● | ● | ● |
| LPO | Ile Grande | RAMCES | 48.80 | -3.58 | 20 | 10 | D | ● | ● | ● | ● |
| GIF | Gif sur Yvette | InGOS/LSCE[11] | 48.71 | 2.15 | 160 | 7 | I | ● | | ● | ● |
| TRN1 | Trainou | InGOS/LSCE[11] | 47.96 | 2.11 | 131 | 5 | I | | | | |
| TRN2 | | | | | | 50 | I | | | | |
| TRN3 | | | | | | 100 | I | | | | |
| TRN4 | | | | | | 180 | I | | ● | ● | ● |
| SCH | Schauinsland | InGOS/UBA[12] | 47.91 | 7.91 | 1205 | 8 | I | ● | ● | ● | ● |
| HPB | Hohenpeissenberg | NOAA | 47.80 | 11.01 | 985 | 5 | D | ● | ● | ● | |
| HUN | Hegyhátsál | InGOS/HMS[13] | 46.95 | 16.65 | 248 | 96 | I | ● | ● | ● | ● |
| HUN | | NOAA | 46.95 | 16.65 | 248 | 96 | D | ● | ● | ● | |
| JFJ | Jungfraujoch | InGOS/EMPA[14] | 46.55 | 7.98 | 3575 | 5 | I | ● | ● | ● | |
| IPR | Ispra | InGOS/JRC[15] | 45.81 | 8.63 | 223 | 15 | I | | ● | ● | ● |
| PUY | Puy de Dome | InGOS/LSCE[11] | 45.77 | 2.97 | 1465 | 10 | I | | ● | ● | ● |
| PUY | | RAMCES | 45.77 | 2.97 | 1465 | 10 | D | ● | ● | ● | |
| BSC | Black Sea | NOAA | 44.17 | 28.68 | 0 | 5 | D | | | | |
| PDM | Pic du Midi | RAMCES | 42.94 | 0.14 | 2877 | 10 | D | ● | ● | ● | |
| BGU | Begur | RAMCES | 41.97 | 3.23 | 13 | 2 | D | ● | ● | ● | |
| LMP | Lampedusa | NOAA | 35.52 | 12.62 | 45 | 5 | D | ● | ● | ● | |
| FIK | Finokalia | RAMCES | 35.34 | 25.67 | 150 | 15 | D | | ● | ● | |

[1] Norwegian Institute for Air Research, Norway

[2] Finnish Meteorological Institute, Helsinki, Finland

[3] Main Geophysical Observatory, St. Petersburg, Russia
[4] University of Edinburgh, Edinburgh, UK
[5] Center for Isotope Research, Groningen, Netherlands
[6] University of Bristol, Bristol, UK
[7] Max Planck Institute for Biogeochemistry, Jena, Germany
[8] Energy research Centre of the Netherlands, Petten, Netherlands
[9] Institut für Umweltphysik, Heidelberg, Germany
[10] University of Science and Technology, Krakow, Poland
[11] Laboratoire des Sciences du Climat et de l' Environnement, Gif-sur-Yvette, France
[12] Umweltbundesamt Germany, Messstelle Schauinsland, Kirchzarten, Germany
[13] Hungarian Meteorological Service, Budapest, Hungary
[14] Swiss Federal Laboratories for Materials Science and Technology, Dübendorf, Switzerland
[15] European Commission Joint Research Centre, Ispra, Italy

**Table 2:** $CH_4$ inversions

| inversion | a priori emissions | period | InGOS station | NOAA+RAMCES discrete air samples |
|---|---|---|---|---|
| S1 | EDGARv4.2FT-InGOS | 2006-2012 | base | ● |
| S2 | EDGARv4.2FT-InGOS | 2010-2012 | extended | ● |
| S3 | no detailed a priori inventory[1] | 2010-2012 | extended | ● |
| S4 | EDGARv4.2FT-InGOS | 2010-2012 | extended | - |

[1] see section 3.1

**Table 3:** Atmospheric models

| Model | Institution | Resolution of transport model: Horizontal (lon × lat) | Vertical | Meteorology | Background $CH_4$ (regional models) |
|---|---|---|---|---|---|
| TM5-4DVAR | EC JRC | Europe: 1° × 1° <br> Global: 6° × 4° | 25 | ECMWF ERA-INTERIM | |
| TM5-CTE | FMI | Europe: 1° × 1° <br> Global: 6° × 4° | 25 | ECMWF ERA-INTERIM | |
| TM3-STILT | MPI-BGC | Europe: 0.25° × 0.25° (STILT) <br> Global: 5° × 4° (TM3) | 61 (STILT) <br> 26 (TM3) | ECMWF operational analysis (STILT) <br> ECMWF ERA-INTERIM (TM3) | TM3[5] |
| LMDZ | LSCE | Europe: ~1.2° × 0.8° <br> Global: ~ 7° × 3.6° | 19 | Nudged to ECMWF ERA-INTERIM | |
| NAME | Met Office | 0.5625° × 0.375° [1] <br> 0.3516° × 0.2344° [2] | 31[3] <br> 59[4] | Met Office Unified Model (UM) | based on measurements at Mace Head[6] |
| CHIMERE | LSCE | 0.5° × 0.5° | 29 | ECMWF ERA-INTERIM | LMDZ[6] |
| COMET | ECN | 0.17° × 0.17° | 60 | ECMWF ERA-INTERIM | TM5-4DVAR |

[1] for simulation period 01/2006-03/2010
[2] for simulation period 03/2010-12/2012
[3] for simulation period 01/2006-10/2009
[4] for simulation period 10/2009-12/2012
[5] coupling based on the method of *Rödenbeck et al.* [2009],
[6] further optimized in the inversion

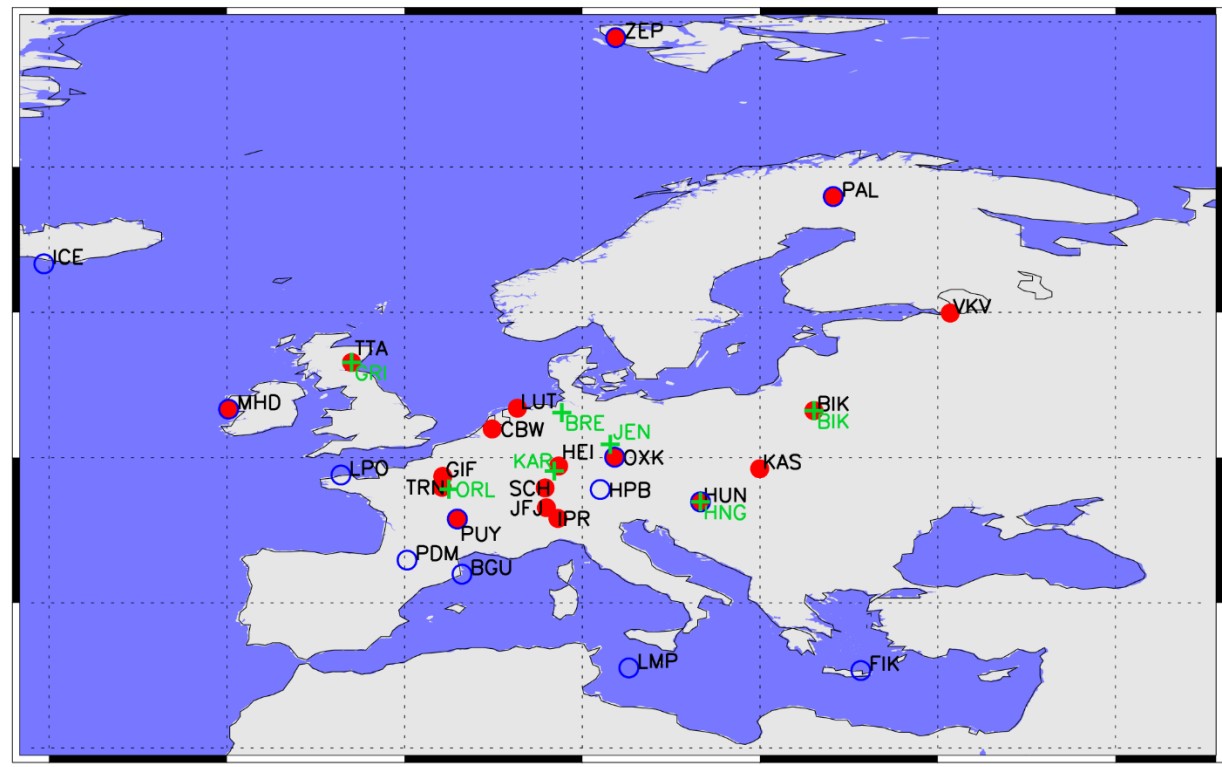

35  **Figure 1**: Map showing locations of InGOS atmospheric monitoring stations with in-situ $CH_4$ measurements (filled red circles), additional stations with discrete air sampling (open blue circles), and the locations of the aircraft profiles (green symbols).

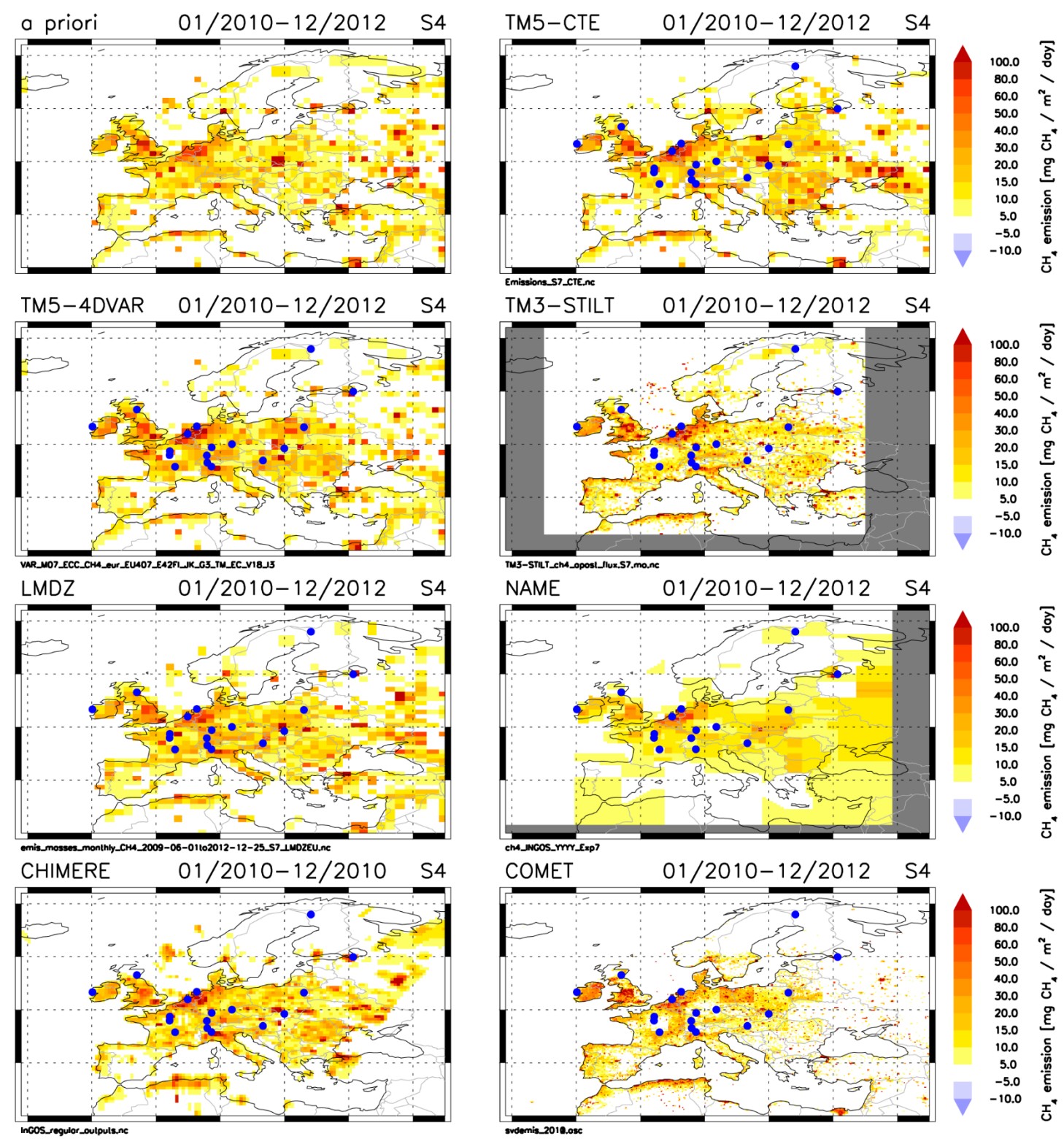

**Figure 2:** European CH₄ emissions derived from the seven inverse models (inversion S4; average 2010–2012; for CHIMERE only 2010). Filled blue circles are the locations of the InGOS measurement stations. Upper left panel shows a priori CH₄ emissions (as applied in TM5-4DVAR at 1°×1° resolution, while regional models use higher resolution for the a priori emissions).

35

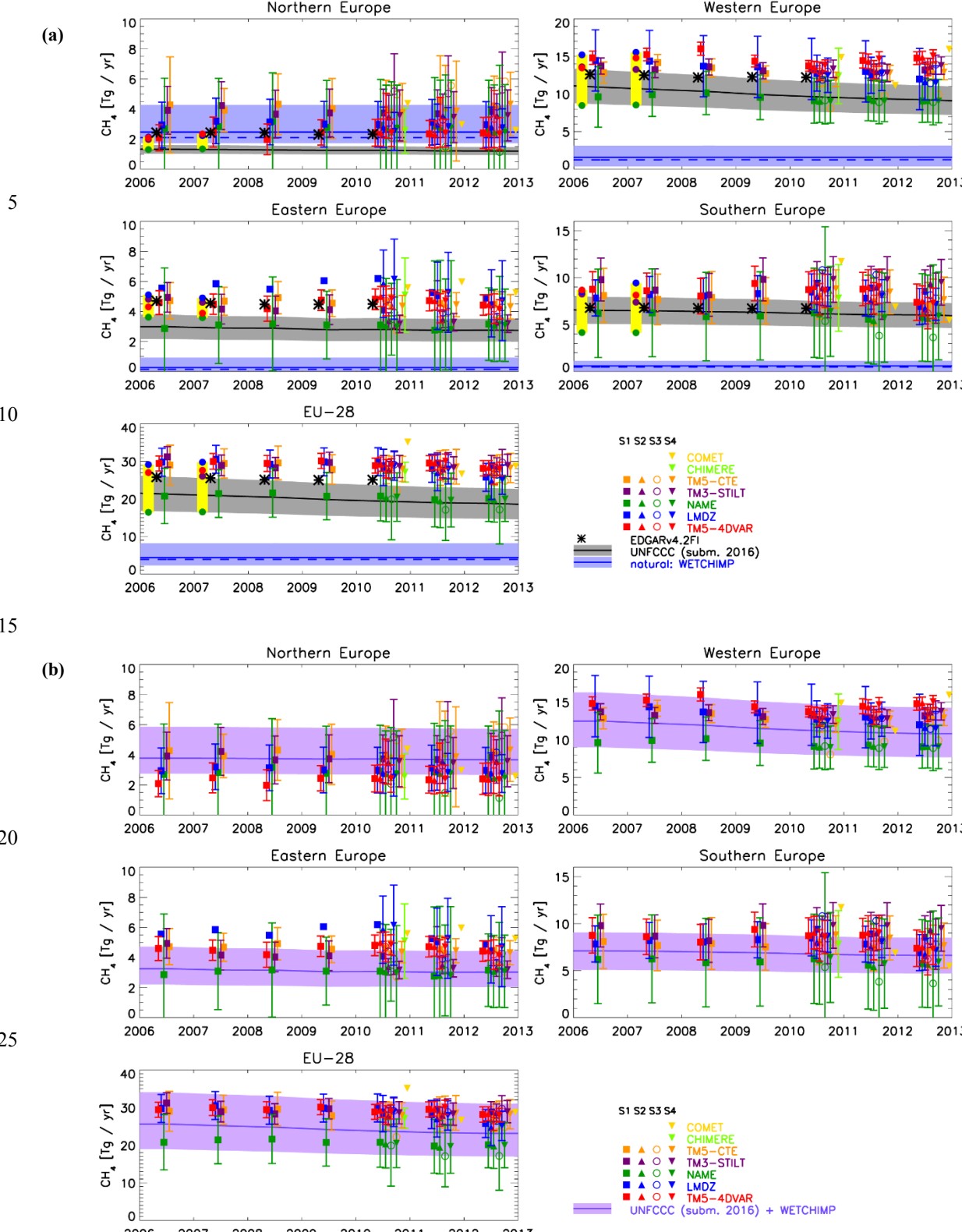

**Figure 3: (a)** Annual total CH₄ emissions derived from inversions for northern, western, eastern, and southern Europe, and for EU-28 (coloured symbols; bars show estimated 2σ uncertainties). For comparison, anthropogenic CH₄ emissions reported to UNFCCC (black line; grey range: 2σ uncertainty estimate based on National Inventory Reports), and from EDGARv4.2FT-InGOS (black stars) are shown. Furthermore, the blue lines show wetland CH₄ emissions from the WETCHIMP ensemble of seven models (mean (blue solid line); median (blue dashed line); minimum-maximum range (light-blue range)). The previous estimates of total CH₄ emissions from *Bergamaschi et al.* [2015] for 2006 and 2007 are shown within the yellow rectangles. **(b)** Comparison of annual total CH₄ emissions derived from inversions with the sum of anthropogenic CH₄ emissions reported to UNFCCC and wetland CH₄ emissions from the WETCHIMP ensemble (violet line; the light-violet range is the combined uncertainty range based on the 2σ uncertainty of UNFCCC inventories and the minimum-maximum range of the WETCHIMP ensemble).

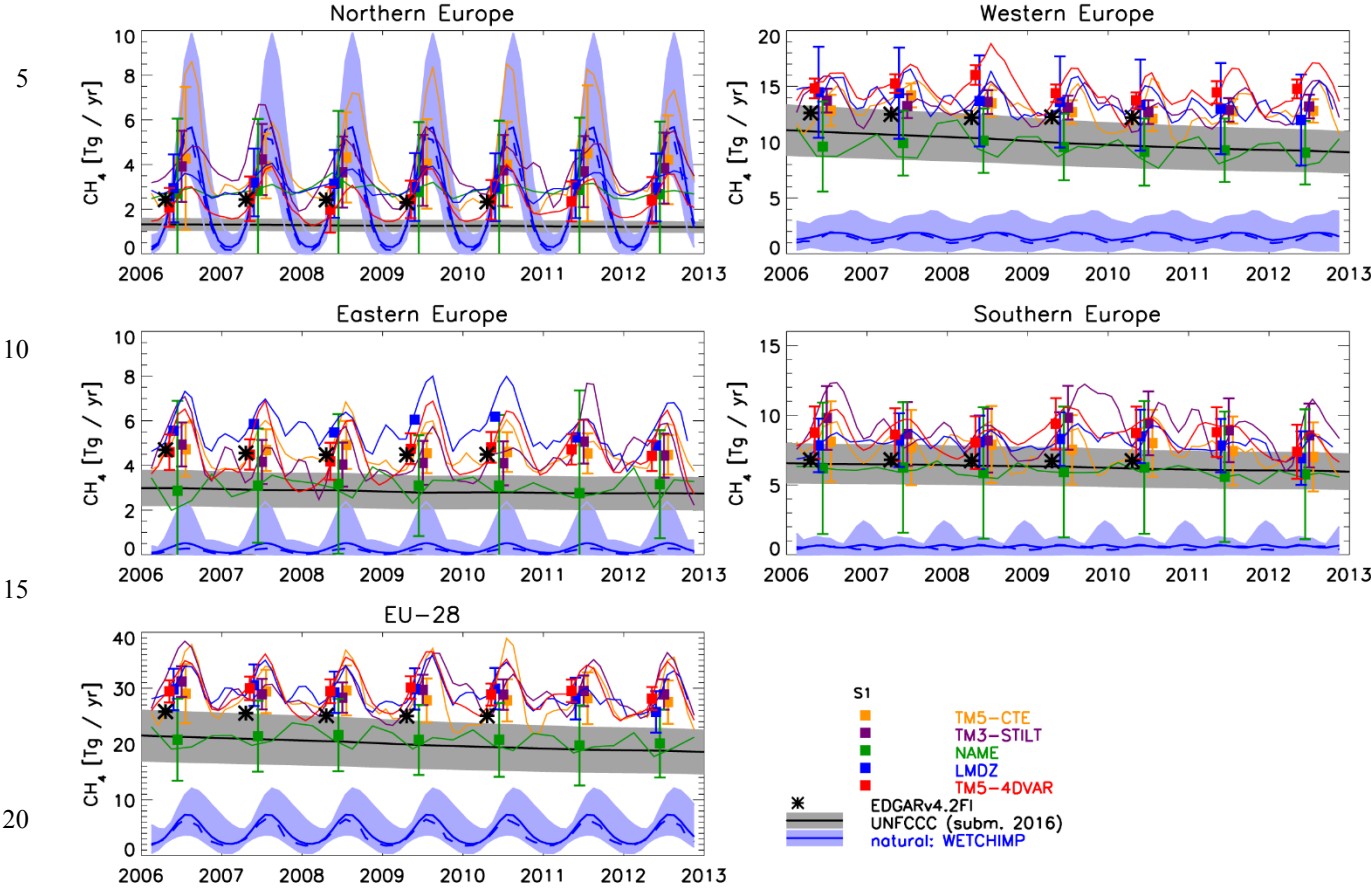

**Figure 4:** Same as Fig. 3a, but including seasonal variation of CH$_4$ emissions derived from the inversions (S1 only; 3-monthly running mean (coloured solid lines)), and seasonal variation of wetland CH$_4$ emissions from the WETCHIMP ensemble of seven models (mean (blue solid line); median (blue dashed line); minimum-maximum range (light-blue range); 3-monthly running mean).

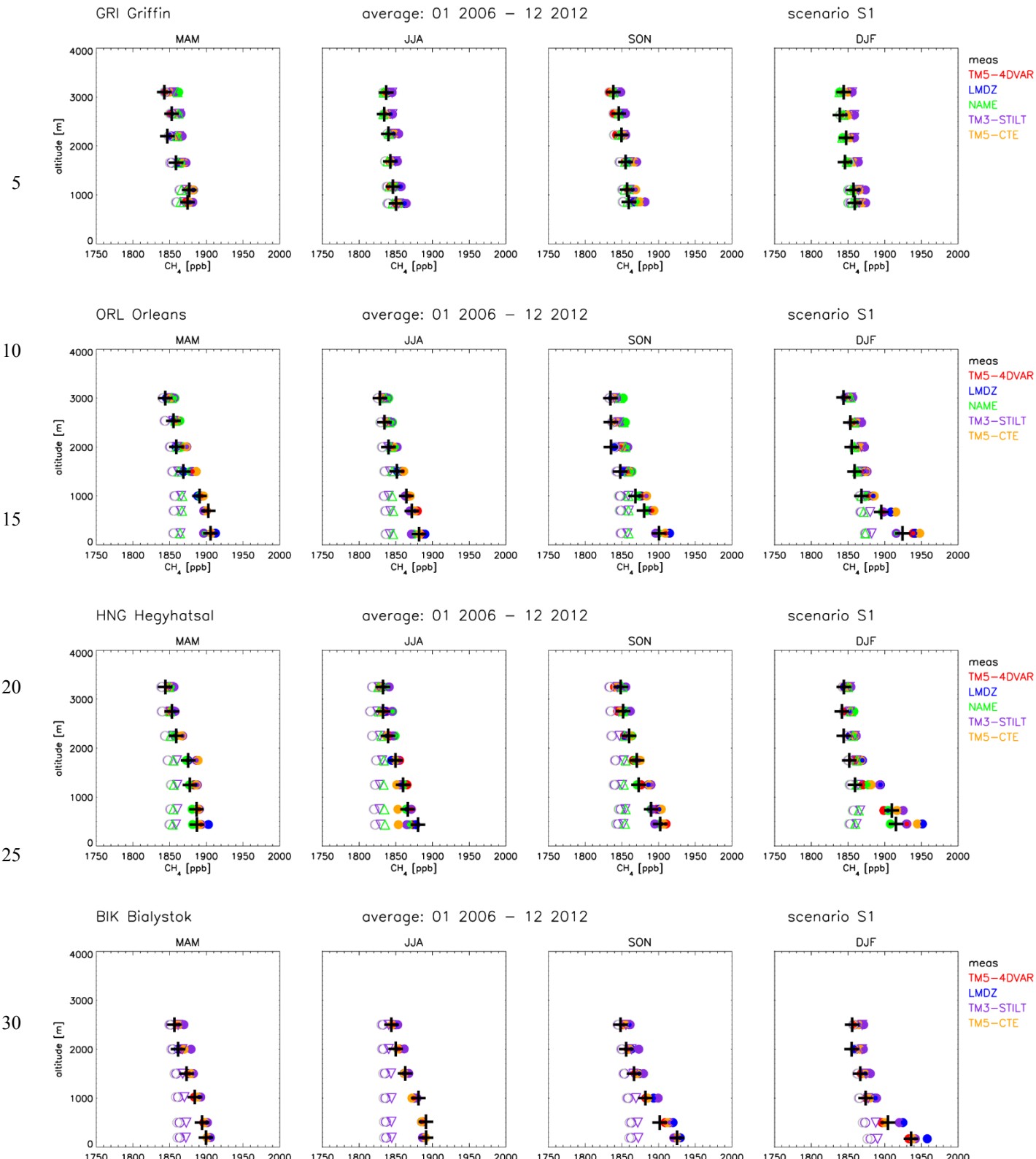

**Figure 5:** Seasonal averages over all available aircraft profile measurements of $CH_4$ at Griffin (Scotland), Orléans (France), Hegyhátsál (Hungary), and Bialystok (Poland) (black crosses) during 2006–2012 and average of corresponding model simulations (filled coloured symbols). The open circles show the calculated background mole fractions, based on the method of *Rödenbeck et al.* [2009], calculated with TM5-4DVAR for the TM5-4DVAR zoom domain (grey), and for the NAME (green) and TM3-STILT (violet) domains (the latter are, however, only partially visible, since they largely overlap with the background for the TM5-4DVAR zoom domain). The open upper triangles (green) are the background mole fractions used in NAME (based on baseline observations at Mace Head), and open lower triangles (violet) are the background mole fractions used in TM3-STILT (based on TM3 model).

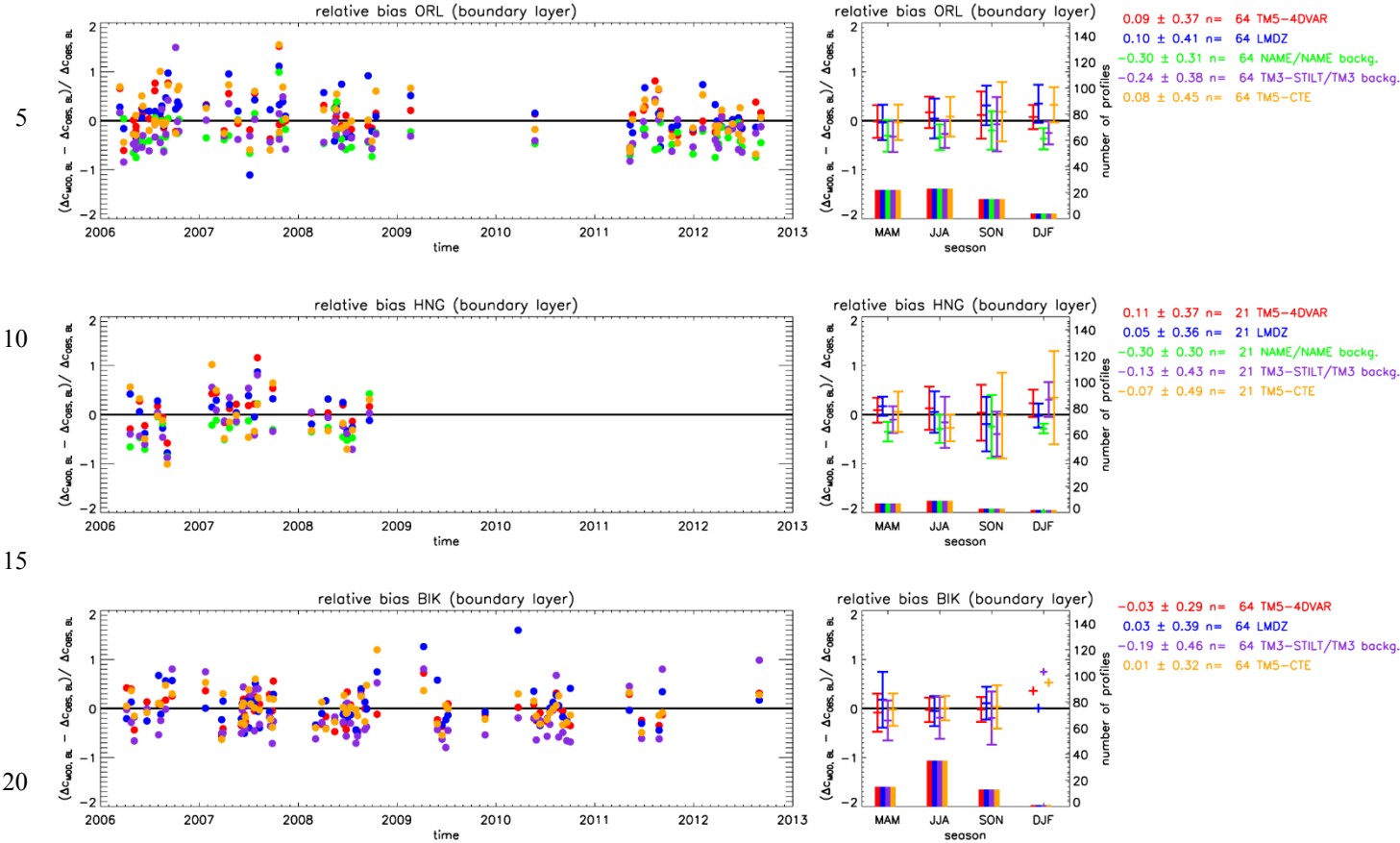

**Figure 6:** 'Relative' bias within the boundary layer evaluated from simulated and observed $CH_4$ mole fraction enhancements compared to the background ($rb_{BL} = (\Delta c_{MOD, BL} - \Delta c_{OBS, BL}) / \Delta c_{OBS, BL}$); see section 4.2). For NAME the model enhancement has been evaluated using the NAME background, for TM3-STILT using the TM3 background, while for all other models the TM5-4DVAR background is used. Left: time series; right: seasonal averages (including 1σ standard deviation) with numbers of available profiles given as bargraphs (see right axis). The numbers on the right side are the average relative bias, 1σ standard deviation, and total number of profiles over the entire period.

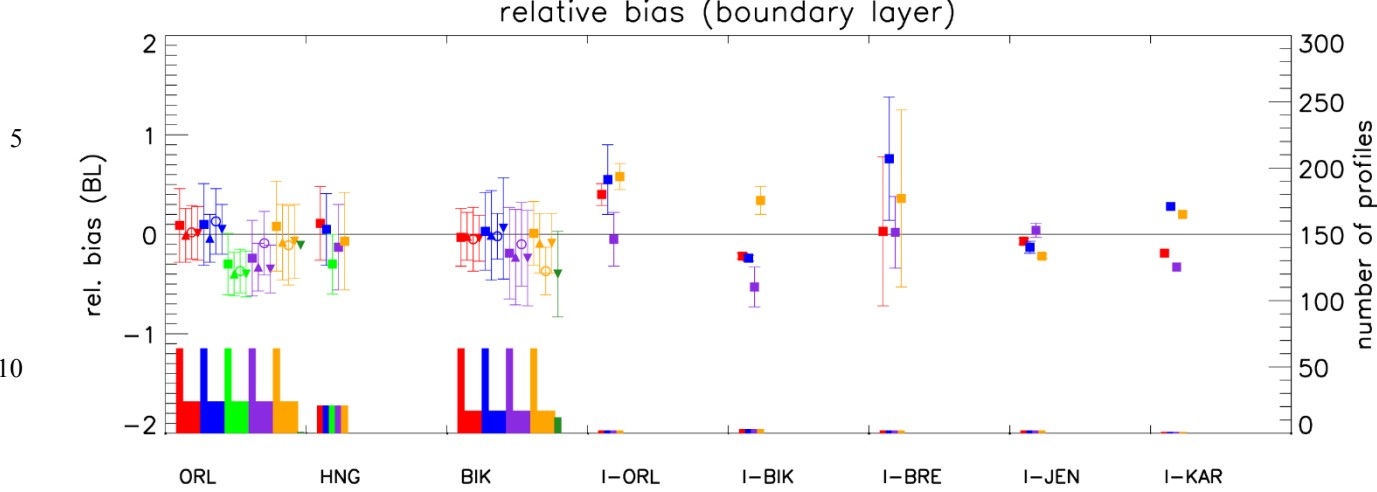

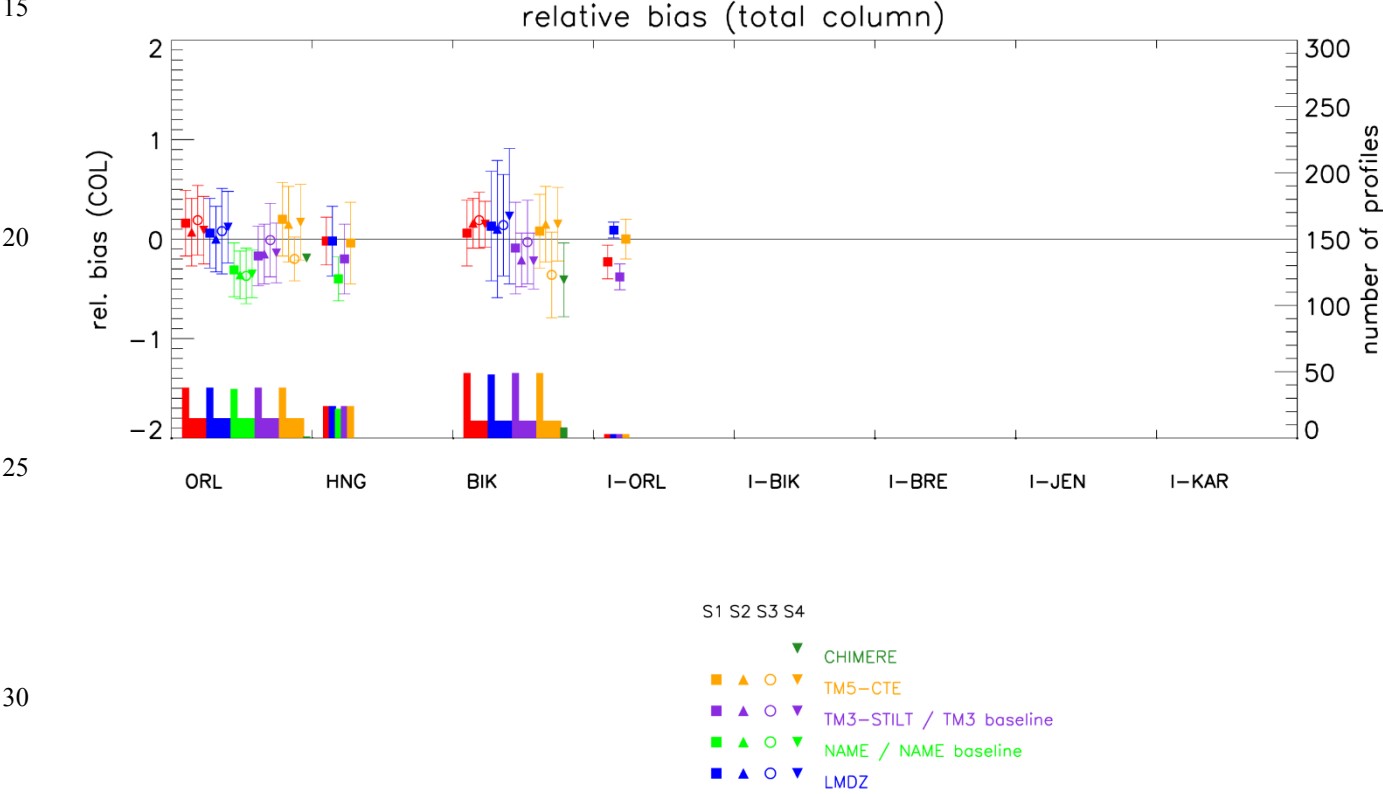

**Figure 7:** Overview of 'relative' bias at different aircraft sites. Top: 'relative' bias within the boundary layer ($rb_{BL}$). Bottom: column-averaged 'relative' bias ($rb_{COL}$). For NAME the relative bias has been evaluated using the NAME background, for TM3-STILT using the TM3 background, while for all other models the TM5-4DVAR background is used. Numbers of available profiles given as bargraphs (see right axis).