# Peer review of "Inverse modelling of European CH4 emissions during 2006-2012 using different inverse models and reassessed atmospheric observations"

_Atmospheric Chemistry and Physics, 2017_

## Editor Comment (EC1)

**Response to re-review of Anonymous Referee #1 and Co-Editor Decision letter for ACP manuscript No.: acp-2017-273 [Peter Bergamaschi et al., Inverse modelling of European CH₄ emissions during 2006-2012 using different inverse models and reassessed atmospheric observations]**

5 This document describes the detailed point-by-point response to re-review of Anonymous Referee #1 and Co-Editor Decision letter by Jens-Uwe Grooß (13 Nov 2017)

(1) *comments from Referee / Co-Editor [in italics]*

(2) **author's response [bold]**

(3) **author's changes in manuscript [bold blue]**.

10

**1. Response to re-review of Anonymous Referee #1**

15 *I do not find the changes between this version and the original manuscript to be compelling enough to change my original stance on this manuscript. It is not clear to this reviewer that this manuscript contributes much to the current literature.*

**We do not agree with this statement, since we had provided a point-by-point reply addressing all points raised by all 3 referees and revised significantly the manuscript.**

20 **We believe that our paper makes a significant contribution to the literature in the field of inverse modelling of European / regional CH₄ emissions and includes many new elements compared to the previous [*Bergamaschi et al., ACP,* 2015] paper:**

**(1) use of the new quality-controlled and harmonized InGOS CH₄ in-situ data set including 18 stations. To our knowledge this is the largest (and most consistent) data set of European atmospheric measurements used**
25 **in any study of top-down estimates of European CH₄ emissions so far.**

**(2) extension of time series (now 7-years period 2006-2012, compared to 2-years period in [*Bergamaschi et al.*, 2015], including a short analysis of CH₄ trends.**

**(3) detailed analysis of potential contribution of natural sources**

**(4) comprehensive validation of model results using an extended set of aircraft observations, providing for**
30 **the first time quantitative estimates of potential biases in derived regional emissions.**

*Point #1 ("I find the wetland hypothesis wholly unconvincing"): The authors have added one sentence to the abstract and one sentence to the conclusions.*

**We did not only include the additional sentences in the abstract and the conclusions, which clearly emphasize the uncertainties of the wetland emissions, especially regarding their spatio-temporal emission distribution (furthermore these uncertainties are discussed in sections 4.1 (already in the original discussion paper)). Following the suggestion of this referee we had included an analysis of the seasonality of derived CH₄ emissions in scenario S3 (which is not using any detailed a priori inventory nor any a priori seasonal cycle; see section 4.1 and new Figure 5S, which confirms that the derived emissions are driven by the observations and not by the a priori inventories). Furthermore, we had addressed the further specific point raised by this referee on this topic, especially the potential impact of the background (see section 4.2, new Figure 14S).**

*Point #2 ("the methods description is poor, making it hard to gain any insight from the different inversions"): The methods description is still poor. There is still just a single paragraph in the main text describing the inversions.*

**As explained in our response to the reviewers, the inverse modelling system are described in the supplementary material (SM), section 1 "Atmospheric models" (summarizing the main elements of each system; this description extends over more than 4.5 pages). Furthermore, all seven inverse models are described comprehensively in separate specific papers (see references in the SM).**

**Since for most models used in this study only smaller updates were applied (compared to previously published applications), we think that is more appropriate to keep the summarizing description of the individual models in the SM.**

**However, following the request of this referee we had updated the description in the SM (e.g. the applied a priori probability density functions (pdf's) for the individual models, and the assumed uncertainties for the observations (including estimates of model errors). Furthermore, we had included the applied boundary conditions (background) in Table 3 and the information about the optimization of the background in the revised main paper.**

**Following also the request of the co-editor, we included further model details in the main paper (section 3.2 / table 3), as detailed below (see under "2. Response to Co-Editor Decision letter by Jens-Uwe Grooß (13 Nov 2017)")**

*It is left to the reader to guess at why the inverse models obtain, in some cases, radically different emissions. Figure 2 is a good example of this. The emissions from COMET look totally different from the others (e.g., why is there a source in Northern Poland that isn't in the prior or any other model?). Figure 2 seems to only be mentioned a single time in the manuscript (in the first paragraph of Section 4.1).*

**A significant part of the 'visual' differences in the spatial patterns between different models is related to different spatial resolutions (and, as explained in the text, for NAME related to the averaging of emissions at larger distances from the observations). Integrated over larger areas (e.g. whole EU-28), the models show a remarkable consistency (apart from the generally lower CH₄ emissions derived by NAME). Differences on smaller spatial scales are probably partly due to differences in model transport and different weighting of**

the observations (i.e. different assumptions of model-data mismatch errors), but may reflect to some extent some noise of the inverse modelling systems.

**In order to address this specific new comment of the reviewer, we have added the following short additional paragraph in the manuscript (in section 4.1):**

**"Apart from this specific feature of the NAME model, also some further differences in the spatial patterns derived by the different models are apparent. One example are the relatively high emissions derived by the COMET model in north-western Poland / north-eastern Germany. Such differences on smaller spatial scales are probably partly due to differences in model transport and different weighting of the observations (i.e. different assumptions of model-data mismatch errors), but may reflect to some extent also some noise of the inverse modelling systems."**

*In the author's response they state: "But this [understanding the differences between top-down emissions] is actually not the goal of this study (and would require further specific modelling experiments). The objective of this study is to use the model ensemble to provide more realistic overall uncertainty estimates (from the range of the inverse models) and to evaluate the model performance by validation against independent observations." Was this not already done in the 2015 paper by many of the same authors (Bergamaschi et al., "Top-down estimates of European CH4 and N2O emissions based on four different inverse models", ACP, 2017)? The authors have just added in a few more models and a few more years of data.*

**We assume that the referee refers here to our [*Bergamaschi et al.*, ACP, 2015] (and not 2017) paper. As outlined above we think that the current paper provides many new elements and provides a significantly extended and improved analysis of European CH$_4$ emissions. What the referee calls "just added... a few more years" is a significant extension of the time period (covering now the 7-years period 2006-2012, while in [*Bergamaschi et al.*, 2015] only the 2 years 2006 and 2007 were analyzed).**

*In general, the conclusions drawn do not seem to be in-line with the analysis. For example, the conclusions of this manuscript state (final paragraph): (2) transport models need to be further improved, including their spatial resolution and in particular the simulation of vertical mixing, aren't some of these models finer-resolution then others and include different treatments of vertical mixing? Do these models actually perform better?*

**We had presented detailed conclusions in-line with our analysis in section "5. Conclusions". The referee refers here to our more general conclusions at the end (which are rather recommendations to further improve top-down estimates in the future). In our study the models with higher spatial resolution do not perform better than TM5-4DVAR with resolution of 1x1 degrees (see short discussion at beginning of section 4.2). Nevertheless we consider the further development of high-resolution models essential to further improve the top-down estimates (e.g. [*Henne et al.*, 2016]).**

*Point #3 ("it's not clear to this reviewer that their "novel" approach to estimate bias is actually an advancement"):
Regarding the 3rd point, their "novel" approach to estimate bias is not particularly useful for estimating biases (as
the authors claim). The difference between simulated and measured enhancements is the term that defines the
model-data mismatch in the cost-function. As I showed in my previous review, $c_{obs}$ - $c_{mod}$ $\equiv$ $\Delta c_{obs}$ - $\Delta c_{mod}$. Following*

5  *this, if there were no difference between the simulated and measured enhancement then the inverse model would
not deviate from the prior. Stated another way, if $\Delta c_{obs} = \Delta c_{mod}$ then the top-down emissions would be equal to the
prior.*

**As explained in our response to the reviewers we do not agree with the statement of the referee. The term**

10  **mentioned by the reviewer is indeed part of the cost function, which the inverse modelling systems aim to
minimize. However, this applies to the observations that are actually used (assimilated) in the inversions,
while we are analyzing here independent observations that were not used in the inversion - which is a
common method to validate inverse models. The concept of using independent observations for validation
of inverse models is described in detail in the recent review paper by *Michalak et al.,* [2017] and has been**

15  **applied in many studies (e.g. [*Alexe et al.*, 2015; *Monteil et al.*, 2013; *Houweling et al.*, 2014; *Bergamaschi et
al.,* 2013]). The rationale behind this approach is to analyze, how well the inverse model perform in areas
which are less constrained by the observations. As mentioned in our initial reply to the reviewer, and
explained in section 4.2 of the manuscript, especially the validation of the vertical profiles (against
independent aircraft profiles) is very important, since the inverse models assimilate only surface observation.**

20  **Therefore, potential errors in the vertical mixing of the models can introduce significant biases in the derived
emission.**
**Commonly, however, such comparisons against independent observations are performed to diagnose only
qualitatively, if the inverse models have biases. The novel aspect of our method is that we provide for the
first time quantitative estimates of the derived regional emissions. We had included in the revised version**

25  **also an analysis of the correlation between the derived relative bias (rb$_{BL}$) and regional model emissions
around the aircraft profile sites, which confirms that rb$_{BL}$ can be used to diagnose biases in the regional
model emissions (new Figure 16S).**
**Since we consider Point #3 as absolutely invalid, no changes have been made in the revised version regarding
this point.**

30

**2. Response to Co-Editor Decision letter by Jens-Uwe Grooß (13 Nov 2017)**

*1. It is not clear to me, why one model that seems to be always at the low end of the deduced CH4 emissions: the
Lagrangian NAME model. Is that because of the Lagrangian formulation of the model? Or because the lack of
observation stations?*

35  **The low emissions of the NAME model are likely - at least partly - related to the positive bias in the
background CH$_4$ used for NAME (as discussed in section 4.2 and shown in the figures 14S (for the aircraft
data), since the regional models invert the difference between the observations and the assumed background.**

**In fact, also at most continental atmospheric monitoring stations (which are applied in the inversion), the background used for NAME (and TM3-STILT) is significantly higher (~10 ppb) compared to the TM5-4DVAR background (Figure 15S).**

**The potential impact of the significant positive bias of the background in NAME (and TM3-STILT) on the derived $CH_4$ emissions is also mentioned in the conclusions of the revised version.**

**It would be certainly useful to perform additional tests with the NAME model using the TM5 baselines in order to quantify the impact of the baseline on the emissions. Unfortunately, however, the NAME team was not yet able to perform this additional test, since they had no further resources after the end of the InGOS project.**

**We don't see any obvious reason why the Lagrangian models should yield lower emissions compared to Eulerian models (however, to our knowledge, this has not been investigated in any study in a systematic way). In fact, also STILT is a Lagrangian model and yields similar emissions as the global / regional Eulerian models.**

**Regarding the observations: All models use the same observational data set (however, the details, how the observations are used in the inversions, differ, in particular the assumed model-data mismatch error, and hence the weighting of the individual observations in the different models).**

**Short general description of the applied the assumed model-data mismatch errors has been included in section 3.2 (see below)**

*2. Can you understand that the annual cycle of wetland emissions is reduced in Northern Europe with respect to the WETCHIMP study? and opposite than the annual cycle increases in the three other parts? Would you say that the WETCHIMP study is incorrect in that respect?*

**We can only speculate about the potential reasons. E.g. the $CH_4$ emissions in the wetland models are highly sensitive to the assumed assumed temperature dependence ($Q_{10}$ values), but also on water table and soil properties (in particular, soil organic carbon content). We note that for Northern Europe the seasonality in the a posteriori emissions derived by TM5-CTE are actually very similar compared to the WETCHIMP mean / median. Nevertheless, it is obvious that the other models derive smaller seasonal cycles in Northern Europe. For the other regions in Europe the derived seasonality is still broadly within the minimum-maximum range of WETCHIMP (even though the seasonality in the mean/median of the WETCHIMP is clearly smaller).**

**We would not say that WETCHIMP is 'incorrect', but clearly the uncertainties are very large, as reflected in the very different spatio-temporal emission patterns of the different individual WETCHIMP inventories.**

*To Point #2 of the review:*

*Although the model descriptions are updated in the supplement, I would likely ask you to add some more information to section 3.2. /table 2, such that in the paper the model diversity would be better understandable*

*without reading the supplement. That should be a few more sentences about the methods (Lagrangian/Eulerian)*
*underlying data (if different) resolution etc.*

**We included further details in section 3.2 / table 3 of the main paper**

**- short description of the applied meteorological fields in the text**

5 **- short description of the applied inverse modelling technique (added both in the text and table 3)**

**- type of model (Eulerian / Lagragian) has now been added also in table 3 (was already in the text)**

**- short general description of the applied uncertainties of the observations (including the measurement and model uncertainties)**

**(model resolutions were already described in the text and included in table 3)**

10

*technical corrections:*

*line 11 4.3 (2.3-8.2) Tg CH4 yr -1*

**'Tg' has been added**

*line 26 globally averaged tropospheric CH4 mole fraction*

15 **'tropospheric' has been added**

**(although this is not exactly the average over the entire troposphere - but with the explanation following in the text ('global average from marine surface sites') it should be clear.)**

20

**References**

Alexe, M., et al., Inverse modelling of $CH_4$ emissions for 2010-2011 using different satellite retrieval products from GOSAT and SCIAMACHY, Atmos. Chem. Phys., 15, 113–133, doi: 10.5194/acp-15-113-2015, 2015.

Bergamaschi, P., et al., Atmospheric $CH_4$ in the first decade of the 21st century: Inverse modeling analysis using
25    SCIAMACHY satellite retrievals and NOAA surface measurements, J. Geophys. Res., 118, 7350–7369, doi: 10.1002/jgrd.50480, 2013.

Bergamaschi, P., et al., Top-down estimates of European $CH_4$ and $N_2O$ emissions based on four different inverse models, Atmos. Chem. Phys., 15, 715-736, doi: 10.5194/acp-15-715-2015, 2015.

Henne, S., D. Brunner, B. Oney, M. Leuenberger, W. Eugster, I. Bamberger, F. Meinhardt, M. Steinbacher, and L.
30    Emmenegger, Validation of the Swiss methane emission inventory by atmospheric observations and inverse modelling, Atmos Chem Phys, 16(6), 3683-3710, doi: 10.5194/acp-16-3683-2016, 2016.

Houweling, S., et al., A multi-year methane inversion using SCIAMACHY, accounting for systematic errors using TCCON measurements, Atmos. Chem. Phys., 14, 3991–4012, doi: 10.5194/acp-14-3991-2014, 2014.

Michalak, A. M., Randazzo, N. A., and Chevallier, F.: Diagnostic methods for atmospheric inversions of long-lived
35    greenhouse gases, Atmos. Chem. Phys., 17, 7405-7421, https://doi.org/10.5194/acp-17-7405-2017, 2017.

Monteil, G., et al., Comparison of $CH_4$ inversions based on 15 months of GOSAT and SCIAMACHY observations, J. Geophys. Res. Atmos., 118, doi:10.1002/2013JD019760, 2013.

---

## Referee Comment (RC2) · Anonymous Referee #2 · 12 Jun 2017

This study presents a multi-model top-down assessment of European methane emissions using the European measurements network. As mentioned, these measurements are performed with the aim to verify bottom-up inventories reported to the UN-FCCC. As such this study can be seen as an assessment of where we are in this process, extending the number of years that were reported in a previous assessment. The results highlight the importance of taking into account natural emissions of methane. Combining natural and anthropogenic emissions the reported total for EU-28 ends up in close agreement with the inventories. The study is a useful reference, and as such it

makes a good contribution to ACP. However, as will be explained below, it also misses some useful opportunities to add value to the previous assessment with the potential to substantially increase the significance of this work. Having gone through the major effort of organizing this model inter-comparison already, the points listed under 'discussion' should receive serious consideration in my opinion.

DISCUSSION

In the context of emission verification, testing the EU-28 total is relevant, however, the network probably resolves additional independent pieces of information. The question is how many, and what this means for the capacity of the European network to resolve country scale emissions. This applies not only to average emissions, but also to their trends. One may argue that in the framework of the COP21 climate agreement the ability to evaluate trends is even more important than the average. Looking at the results that are presented, information about trends is clearly visible in the time series, but to my surprise it is not discussed at all. Even if it turns out that these trends are not significant it is useful to quantify and discuss how far we are from this target. It is a bit surprising that the multi-year time dimension, which is the new element of this study compared to the previous one, is left unexplored.

A useful attempt is made to assess biases in transport models using vertical profile measurements. However, what is missing is the link between these biases and the inverted emissions. It is mentioned that those models that overestimate PBL average CH4 should overestimate emissions. In fact, all the ingredients are available to quantify this link and assess the impact of transport biases on emissions. It raises the question why this is not done. Is it an important factor explaining the range of emission that are found or not?

SPECIFIC COMMENTS

page 4, line 6: Which targets are set by the quality control mentioned here? Are they met?

page 5, line 16: Using constant a priori flux uncertainties also? How do these emissions/uncertainties relate to those of the other scenarios?

page 5, line 24: Do the regional models (apart from NAME) prescribe boundary conditions, or allow further optimization?

page 8, line 10-15: It would be good to mention some typical numbers here for the bottom up and top down derived seasonal amplitudes (it is not so clear to see from figure 4)

page 8, line 30-35: How about the seasonality in the energy sector? (domestic heating etc.)

page 9, line 7: The difference between the observed vs simulated amplitude of variability (as used in Taylor diagrams for instance) provides a piece of information that is more independent from correlation as the RMS that is used here.
* * *

---

## Referee Comment (RC3) · Anonymous Referee #3 · 13 Jun 2017

Summary/General comments: The manuscript presents 'top-down' optimized methane emissions for Europe for the 2006-2012 time period. A new, harmonized 18 site-monitoring network is used with seven inverse models and four experiments. Optimized emissions are reported (and are overall consistent between top-down and bottom-up), biases are assessed using aircraft data, and the inference of a non-negligible wetland source is intimated. Overall it is interesting and important work to pursue. It is not easy to use this many different model/inverse approaches to one regional question, and this can potentially provide substantially more information and understanding for

how to best quantify fluxes with atmospheric observations. This paper is well-placed in ACP. However, there are a couple important gaps that need to be addressed before I can recommend publication. Most importantly, the description of different models and inverse methods is somewhat lacking, this should be a central element of this work, and this needs to be improved before I can recommend publication.

Major comments: Models/Inverse methods: There is limited discussion of the different models, and specifically, of the inverse methodology being employed by each model. I understand much of this is referenced to various previous publications, and the supplement does go through each model independently, but it is important for the reader to see more comparative details in this manuscript to be able to understand the differences between models/inversions and possible nuanced causes. A succinct but clear description in its own section of the different inverse approaches used and the subtle "expert-user" choices made to define the inversion would be essential. For example the prior uncertainties and correlations lengths, which are defined differently in the different inversions, could be rather impactful on the results. How were these different priors chosen, and how important is this choice? The authors have conducted multiple experiments – they need to better convey to the reader the differences between the inversions and experiments so we can better assess the meaning of similar/different results. In many ways this could be one of the biggest contributions of this paper.

Sensitivity of network to domain: Western Europe has the highest density of observation sites, and measurement density (and sensitivity to emissions) falls off rapidly in other regions of Europe. Given this, how appropriate is it to lump the entirety of the domain together? I'd like to see a little more discussion of the sensitivity of the network and therefore dependence of prior/assumptions in some of the domains. Another way to consider this question is how many regions can the network distinguish, and how do these regions compare with geopolitical domains? This impacts my next point.

Importance of wetlands: I'm not sure if from this analysis alone the authors can conclude substantial wetland source are or are not required to match observations.

The largest prior wetland estimate (and seasonality) is in Northern Europe, where there are few observation points and the inverted seasonality is actually smaller than WETCHIMP models. When aggregating all of Europe together, it would appear the added emissions and seasonality from WETCHIMP is helpful in bringing bottom-up and top-down closer together – but given this point of spatial/seasonal errors in the Northern Europe domain I'm not sure this overall improvement is indicative of a better representation or coincidence where the inversion finds large seasonality in other regions of Europe where WETCHIMP models do not expect significant wetland sources. I would think the authors should tone done the statement of wetlands importance in the abstract, and also would like to see further defense of the seasonality signal observed and attribution that it must be wetlands.

---

## Author Comment (AC2) · 13 Jul 2017

*This study presents a multi-model top-down assessment of European methane emissions using the European measurements network. As mentioned, these measurements are performed with the aim to verify bottom-up inventories reported to the UN-FCCC. As such this study can be seen as an assessment of where we are in this process, extending the number of years that were reported in a previous assessment. The results highlight the importance of taking into account natural emissions of methane. Combining natural and anthropogenic emissions the reported total for EU-28 ends up*

[Figure]

*in close agreement with the inventories. The study is a useful reference, and as such it makes a good contribution to ACP. However, as will be explained below, it also misses some useful opportunities to add value to the previous assessment with the potential to substantially increase the significance of this work. Having gone through the major effort of organizing this model inter-comparison already, the points listed under 'discussion' should receive serious consideration in my opinion.*

We thank the reviewer for the very positive overall evaluation of our study.

**DISCUSSION**

*In the context of emission verification, testing the EU-28 total is relevant, however, the network probably resolves additional independent pieces of information. The question is how many, and what this means for the capacity of the European network to resolve country scale emissions. This applies not only to average emissions, but also to their trends. One may argue that in the framework of the COP21 climate agreement the ability to evaluate trends is even more important than the average. Looking at the results that are presented, information about trends is clearly visible in the time series, but to my surprise it is not discussed at all. Even if it turns out that these trends are not significant it is useful to quantify and discuss how far we are from this target. It is a bit surprising that the multi-year time dimension, which is the new element of this study compared to the previous one, is left unexplored.*

The anthropogenic CH4 emissions reported to UNFCCC have indeed decreased between 2006 and 2012 by 11.6%. The models show rather smaller trends (which are in most cases indeed probably not significant). An evaluation of the uncertainties of the trends, however, is very difficult, since this requires information about the error correlations between subsequent years (which is not available). We will include a short discussion of the trends in the revised version.

*A useful attempt is made to assess biases in transport models using vertical profile measurements. However, what is missing is the link between these biases and the*

[Figure]

*inverted emissions. It is mentioned that those models that overestimate PBL average CH4 should overestimate emissions. In fact, all the ingredients are available to quantify this link and assess the impact of transport biases on emissions. It raises the question why this is not done. Is it an important factor explaining the range of emission that are found or not?*

Following the suggestion of the reviewer we analyzed the relationship between the estimated relative bias (based on the enhancement compared to the background integrated over the boundary layer) and the model emissions in the area around the regular aircraft profiles sites. The analysis showed significant correlations between model emissions and estimated model bias. We will include this analysis in the revised version.

**SPECIFIC COMMENTS**

*page 4, line 6: Which targets are set by the quality control mentioned here? Are they met?*

No specific threshold values have been set. The typical range for the "working standard repeatability" is ∼1-4 ppb. Since this "working standard repeatability" is used by the inverse models, measurements with higher "working standard repeatability" are weighted less in the inversion.

*page 5, line 16: Using constant a priori flux uncertainties also? How do these emissions / uncertainties relate to those of the other scenarios?*

For inversion S3 very large uncertainties of the homogeneous a priori fluxes were assumed (ranging between 200% and 500% per grid-cell and month; see model description in the supplementary material) in order to give the inversion enough degree of freedom to retrieve regional emission hot spots (which have much higher emissions than the applied homogeneous a priori fluxes). In contrast, the assumed uncertainties per grid cell and months are much smaller for the other scenarios (typically 100%).

*page 5, line 24: Do the regional models (apart from NAME) prescribe boundary conditions, or allow further optimization?*

Apart from NAME, the boundary conditions are further optimized also in CHIMERE, while the other regional models used prescribed boundary conditions. These boundary conditions were derived from optimized concentrations of global inversion systems (STILT: from TM3, COMET: from TM5-4DVAR, CHIMERE: from LMDZ).

*page 8, line 10-15: It would be good to mention some typical numbers here for the bottom up and top down derived seasonal amplitudes (it is not so clear to see from figure 4)*

We will add the numbers of the derived seasonal amplitudes in the revised version.

*page 8, line 30-35: How about the seasonality in the energy sector? (domestic heating etc.)*

No or only small seasonal variations were found in the limited number of studies investigating natural gas distribution system [Wennberg et al., 2012; McKain et al., 2014]. Wong et al. [2016] argued that "the natural gas distribution pipeline system is pressure-regulated at several points, and leakage should be independent of consumption to first order", but that natural gas storage facilities may have seasonally varying leakage rates, depending on energy demands.

*page 9, line 7: The difference between the observed vs simulated amplitude of variability (as used in Taylor diagrams for instance) provides a piece of information that is more independent from correlation as the RMS that is used here.*

Following the suggestion of the reviewer we will analyze also the difference between the observed vs simulated amplitude of variability.

**References**

McKain, K., et al., Methane emissions from natural gas infrastructure and use in the

urban region of Boston, Massachusetts, PNAS, 112, 1941–1946, 2015.

Wennberg, P. O., et al., On the Sources of Methane to the Los Angeles Atmosphere, Environmental Science  Technology, 40 46(17), 9282-9289, doi: 10.1021/es301138y, 2012.

Wong, C. K., T. J. Pongetti, T. Oda, P. Rao, K. R. Gurney, S. Newman, R. M. Duren, C. E. Miller, Y. L. Yung, and S. P. Sander, Monthly trends of methane emissions in Los Angeles from 2011 to 2015 inferred by CLARS-FTS observations, Atmos. Chem. Phys., 16(20), 13121-13130, doi: 10.5194/acp-16-13121-2016, 2016.

---

## Author Response (AR1)

Response to all referee comments for ACP manuscript No.: acp-2017-273 [Peter Bergamaschi et al., Inverse modelling of European CH4 emissions during 2006-2012 using different inverse models and reassessed atmospheric observations]

- 5 This document describes the detailed point-by-point response to all three referee comments
  - (1) comments from Referees [in italics]
  - (2) author's response [regular fonts]
  - (3) author's changes in manuscript [blue].

Furthermore, additional updates of the manuscript are described in section 2

**10**

**1. Response of referee comments**

**1.1 Anonymous Referee #1**

**Major comments - "2.1 Wetland hypothesis"**

I do not find these arguments convincing. The arguments, as presented, are inconclusive at best. The region
 where we would expect the largest wetland emissions is Northern Europe, however in this region the inversions
 consistently point to a reduced seasonal cycle compared to WETCHIMP. The EU-28 seasonal cycle in WETCHIMP
 is~10 Tg/yr which is roughly the same as the top-down seasonal cycle in their inversions. But, again, their
 inversion pointed to a decrease in the seasonal cycle in Northern Europe where the bulk of the wetland emissions
 should be. So why do we think this is due to wetlands? Because other sources are assumed to be atemporal? The

20 authors acknowledge that other sources could have seasonal cycles (e.g., manure emissions are temperature dependent, enteric fermentation could have a seasonal cycle due to variations in the herd size, etc).

Although the WETCHIMP model ensemble estimates large CH4 emissions for Northern Europe (1.9 (0.8-3.5) Tg CH4 yr-1 (mean, minimum, maximum); excluding Norway), this data set estimates significant wetland emissions also for western Europe (1.6 (0.4-3.1) Tg CH4 yr-1), eastern Europe (0.3 (0.03-0.9) Tg CH4 yr-1) and southern

25 Europe (0.6 (0.01-1.1) Tg CH4 yr-1). Excluding Northern Europe, the sum of the WETCHIMP CH4 emissions for western, eastern, and southern Europe is 2.5 (0.4-5.1) Tg CH4 yr-1, corresponding to 12.5% (2.2%-25.6%) of the total reported anthropogenic CH4 emissions for EU-28, which highlights the potential significant contribution of wetland emissions also for western / eastern / southern Europe.

While the inversions of TM5-4DVAR, TM5-CTE, TM3-STILT yield indeed a smaller seasonal cycle for Northern
Europe compared to the mean of the WETCHIMP models (but similar amplitude for TM5-CTE), they derive significant seasonal cycles also for western / eastern / southern Europe, broadly consistent with the range of seasonal variations of the WETCHIMP ensemble. Our interpretation of this result is that indeed the spatial distribution of wetland emissions of the WETCHIMP ensemble (within Europe) is not fully consistent with the inversion results, but we consider the considerable derived seasonal variation for western / eastern / southern
Europe as indication that wetlands could contribute significantly also in these sub-regions.

This interpretation is indeed based on the assumption that anthropogenic CH4 emissions have only very small seasonal variations. To our knowledge, only very few studies investigating the seasonal variations of the

anthropogenic emissions are available (and have been discussed in the discussion paper). Clearly further studies on this topic will be required.

We will emphasize more clearly in the revised paper the caveats of the hypothesis of significant wetland emissions.

- 5 We emphasize now more clearly the caveats in the abstract ("However, the contribution of natural sources remains rather uncertain, especially their regional distribution.") and the conclusions ("Furthermore, it needs to be emphasized that wetland inventories have large uncertainties and show large differences in the spatial distribution of CH4 emissions.")
- 10 There is little-to-no discussion of the background used for the region (see next comment), could errors in the background be driving this?

The global models assimilate also global observations from the NOAA ESRL global cooperative air sampling network. The model simulations outside Europe have been further analyzed for TM5-4DVAR, showing in general very good agreement with observations at global background stations (similar as shown in previous papers, see

15 Bergamaschi et al. [2013], Figure S4a). Therefore, it seems unlikely, that errors in the background are driving the derived seasonal variations of European CH4 emissions.

We have extended the analysis of the background (see also "2. Further updates of the manuscript"), and evaluated the quality of the background by comparing model simulations with the aircraft observations for events with very low simulated contribution ( $\leq$  3 ppb) from European CH4 emissions (new Fig. 14S). This

- 20 analysis suggests that the background calculated by TM5-4DVAR is relatively realistic, while the regional models NAME and STILT have a positive bias in the background. This is now further discussed in section 4.2 ("Evaluation of inverse models"). However, there is no indication for a seasonal bias of the inversions (see updated Figure 6 and discussion in section 4.2).
- 25 There is no mention of the methane sink, is the OH correct? If OH were too low then you may have an artificially low seasonal cycle in the global simulations (which would, again, impact the background concentrations).

The global models apply OH fields that were calibrated against methyl chloroform measurements [Patra et al, 2011; Bergamaschi et al., 2010; Houweling et al., 2014]. Since the global models assimilate global observations, potential deficiencies of the global OH fields are likely to be largely compensated by (artificial) increments of the

30 global fluxes. As mentioned above, e.g. TM5-4DVAR reproduces the measurements at global background stations very well (the performance of other global model at global sites were not further investigated in this study). The impact of different global OH fields on derived European CH4 emissions has been investigated by Bergamaschi et al. [2010], which showed only a very small impact.

The sensitivity experiment for TM5-4DVAR using different global OH fields [Bergamaschi et al., 2010] has been included in the description of TM5-4DVAR (supplementary material, section 1.1).

It's unclear to this reviewer why the authors did not just perform an inversion with atemporal emissions and compare the posterior seasonality to the prior seasonality. This would show how much of this derived

seasonality comes from the data instead of the prior. It would allow them to say which regions have significant seasonal cycles. The authors could have achieved much of this by looking at the seasonal cycles in their case with homogenous prior emissions.

Also the inversion results from inversion S3 (which was performed without using detailed bottom-up

5 inventories as 'a priori'), show significant seasonal cycles in derived emissions. This confirms that the derived seasonal cycle is driven by the observations, and not by the a priori emissions. This was not mentioned in the discussion paper but will be included in the revised paper.

We have included a new figure (Fig. 5S) with the mean seasonal cycles for all inversions (including S3) and added a short discussion in section 4.1.

**10**

20

**Major comments - "2.2 Poor description of methods makes it difficult to gain any insight"**

The description of the various inversion systems is poor. There is a single paragraph in the main text describing the inversions. There is no mathematical description of the inversions. This is quite surprising since, at it's core, this is an inversion paper.

15 The inverse modelling system are described in the supplementary material (SM), section 1 "Atmospheric models" (summarizing the main elements of each system). Furthermore, all seven inverse models are described comprehensively in separate specific papers (see references in the SM).

For most models used in this study only smaller updates were applied (compared to previously published applications). Therefore, we had chosen to put the model descriptions in the SM (and would prefer to keep this in the SM also in the revised version).

However, we will somewhat extend the general description of the models in the main paper (section 3.2 "Atmospheric models") in the revised version.

We have included the applied boundary conditions (background) in Table 3 and included the information about the optimization of the background in the text. Some further details were added (or updated) in the supplementary material

**25 supplementary material.**

**At the bare minimum, the author's should state the assumptions for their inversions (e.g., Gaussian errors?).**

Most inverse modelling systems applied in this study use Gaussian probability density functions for the uncertainties of the emissions (in case of TM5-4DVAR a 'semi lognormal' pdf is used; see SM section 1.1).

30 We will add the applied pdfs in the model description for those models where this information is missing in the discussion paper.

We have added the information about the applied probability density functions (pdfs) in the detailed model description in the supplementary material.

There is additional text in the supplement (~1 paragraph per model) but it is difficult to synthesize the models. Some of the models are regional but it's not clear where the boundary conditions are coming from. It is clearly stated in section 3.2 ("Atmospheric models"; page 5, lines 23-25) where the boundary conditions are coming from:

"The regional models use boundary conditions from inversions of the global models (STILT from TM3, COMET from TM5, CHIMERE from LMDZ, or estimate the boundary conditions in the inversions (NAME), using baseline observations at Mace Head as 'a priori' estimates."

5

Furthermore, the boundary conditions are described also in the SM for all regional models (STILT, NAME, CHIMERE, COMET).

Although already described in the text, we have included now the boundary conditions (background CH4 mole fractions) also in Table 3.

10

**Some of the models are estimating the covariance matrices from the data, some are not.**

We assume that the reviewer refers here to the observation covariance matrix. The uncertainties of the observations (diagonal elements of the covariance matrices) include both the measurement error and the model error. Most models use the "working standard repeatability" (see section 2 of main paper) as observation

error. However the estimates of the model errors are very different in the different inverse modelling systems 15 (and generally based on simplified assumptions). For most models the assumed uncertainties of the observations is described in SM section 1 - for those models where this information has been missing (CHIMERE, COMET), it will be added.

The assumed uncertainties for the observations (including estimates of model errors) are now described for all 20 models in the supplementary material (section 1).

It is extremely difficult for the reader to understand why these inversions are performing differently. For example, it seems that the boundary conditions are coming from global models in the case of some regional models, how independent are these different inversion systems (especially the global/regional ones)? Are we

25 comparing apples to apples?

> The global models providing the boundary conditions for the regional models are generally largely independent from the regional models (apart from the fact that the different models may have some features in common, e.g. use of same or similar meteo data sets).

No change in the manuscript regarding this point.

30

How much of the differences are due to assumptions vs transport vs something else? It's extremely difficult to understand the differences without clearly laying out the key differences between the models.

Given the very high complexity of the different inverse modelling systems, it is indeed very difficult to understand where the differences in the derived emissions are coming from. But this is actually not the goal of

35 this study (and would require further specific modelling experiments). The objective of this study is to use the model ensemble to provide more realistic overall uncertainty estimates (from the range of the inverse models) and to evaluate the model performance by validation against independent observations.

The background mole fractions have been identified as one major parameter which can lead to biases in the derived emissions. This is now discussed in more detail in the revised version (section 4.2, see also "2. Further updates of the manuscript").

5 I would point the authors to the Henne et al. (2016) paper as an example of a paper that does a good job of explicitly highlighting the differences between their inversion systems and allows the readers to actually gain insight from the ensemble of inversions. Table 2 from Henne et al. (2016) is a particularly good example of how one can demonstrate the major differences between inversion frameworks.

The fundamental difference between the study of Henne et al. (2016) and our study is that Henne et al. use one

- single inverse modelling system, varying various input parameters / settings of this system as compiled in their 10 Table 2. In contrast, our study uses very different inverse modelling systems, which makes it inherently more difficult to highlight the differences between the systems (which are largely independent systems and which differ in many aspects). Important parameters (model resolution, meteorology, a priori emission inventories, applied station sets are compiled in Tables 1, 2, and 3. We will include also the applied baselines for the regional
- 15 models in Table 3.

We have included the applied boundary conditions (background) in Table 3.

Also, the phrase "no a priori" is, almost certainly, using incorrect terminology. The posterior probability is proportional to the product of the likelihood and the prior probability: Posterior probability / Likelihood × Prior

20 probability. Using a homogenous distribution of emissions is still including a prior, it just isn't based on a bottomup inventory. To actually use "no a priori" would be "Maximum Likelihood Estimation" where one simply finds the parameters that maximize the likelihood term

In section 3.1 we have described S3 as:

"Inversion S3 was performed without using detailed bottom-up inventories as 'a priori', in order to analyse the 25 constraints of observed atmospheric CH4 on emissions independent of 'a priori' information (using a homogeneous distribution of emissions over land and over the ocean, respectively, as starting point for the inversions in a similar manner as in Bergamaschi et al. [2015])."

The short notion "no a priori" has been only used in Table 2. We will add a footnote in this table to refer the reader to the above description in section 3.1

30 "no a priori" has been changed to "no detailed a priori inventory" and footnote has been added.

**Major comments - "2.3 'Novel' Bias method"**

This "novel" bias method is, essentially, what an inversion already does... They are just plotting the model-data mismatch averaged over different parts of the atmosphere. This is hardly a "novel approach".

35 (mathematical derivation not repeated here)

From this, it's quite easy to see how cobs-cmod =  $\Delta$ cobs- $\Delta$ cmod. So, as I stated above, all the authors have done is plot the model-data mismatch (cobs – cmod) averaged over two parts of the atmosphere. It does not strike this reviewer as particularly "novel".

We do not agree with the statement of the reviewer that our approach to estimate the bias in the derived

6 emissions is "essentially, what an inversion already does", since we look at independent observations that were not used in the inversion - which is a common method to validate inverse models (see e.g. Michalak et al.,
 [2016]). Commonly, however, such analyses are performed to diagnose qualitatively, if the inverse models have biases.

The novel aspect of our method is that we use the baseline in order to extract the signal which comes from the

- 10 European emissions. Integrating the enhancement of the model simulations compared to the background over the entire boundary layer or the entire column of the lower troposphere (and comparison with the corresponding observed CH4 enhancement) provides a measure of the total CH4 emitted by European emission. The ratio of the simulated vs. observed integrated enhancements provides a first order estimate of the relative bias in the model emissions.
- 15 As explained in section 4.2, the validation against independent aircraft profiles is very important, since the inverse models assimilate only surface observation. Therefore, potential errors in the vertical mixing of the models can introduce significant biases in the derived emission.

Independent from the the comment of the reviewer (with which we do not agree), we have updated the analysis of the model biases (see also "2. Further updates of the manuscript").

20

There are novel approaches that attempt to account for systematic errors in inversions in a rigorous manner. Weak-Constraint 4D-Var (Tremolet, 2006) and Hierarchical Bayesian inference (see Ganesan et al., 2014 and references therein) are two good examples of this.

We agree that the "Hierarchical Bayesian inference" is an interesting approach to provide more realistic

25 uncertainty estimates for individual models (i.e. estimates within the individual inverse modelling systems, corresponding to the error bars in our Figure 3). Nevertheless, validation against independent observations will remain indispensable as independent evaluation of the inverse models.

Also the mentioned "Weak-Constraint 4D-Var" is certainly a very interesting technique - but to our knowledge so far only applied in some cases for data assimilations, but not in inverse modelling systems.

**30 No change in the manuscript regarding this point.**

**1.2 Anonymous Referee #2**

This study presents a multi-model top-down assessment of European methane emissions using the European measurements network. As mentioned, these measurements are performed with the aim to verify bottom-up

35 inventories reported to the UNFCCC. As such this study can be seen as an assessment of where we are in this process, extending the number of years that were reported in a previous assessment. The results highlight the importance of taking into account natural emissions of methane. Combining natural and anthropogenic emissions the reported total for EU-28 ends up in close agreement with the inventories. The study is a useful reference, and as such it makes a good contribution to ACP. However, as will be explained below, it also misses some useful opportunities to add value to the previous assessment with the potential to substantially increase the significance of this work. Having gone through the major effort of organizing this model inter-comparison already, the points listed under 'discussion' should receive serious consideration in my opinion.

5 We thank the reviewer for the very positive overall evaluation of our study.

**DISCUSSION**

In the context of emission verification, testing the EU-28 total is relevant, however, the network probably resolves additional independent pieces of information. The question is how many, and what this means for the capacity of the European network to resolve country scale emissions. This applies not only to average emissions,

- 10 but also to their trends. One may argue that in the framework of the COP21 climate agreement the ability to evaluate trends is even more important than the average. Looking at the results that are presented, information about trends is clearly visible in the time series, but to my surprise it is not discussed at all. Even if it turns out that these trends are not significant it is useful to quantify and discuss how far we are from this target. It is a bit surprising that the multi-year time dimension, which is the new element of this study compared to the previous
- 15 one, is left unexplored.

The anthropogenic CH4 emissions reported to UNFCCC have indeed decreased between 2006 and 2012 by 11.6%. The models show rather smaller trends (which are in most cases indeed probably not significant). An evaluation of the uncertainties of the trends, however, is very difficult, since this requires information about the error correlations between subsequent years (which is not available). We will include a short discussion of the trends in the error discussion of the trends.

20 trends in the revised version.

We have included a short discussion of the  $CH_4$  trends at the end of section 4.1 (and included the new Figure 7S in the supplementary material).

A useful attempt is made to assess biases in transport models using vertical profile measurements. However,
 what is missing is the link between these biases and the inverted emissions. It is mentioned that those models that overestimate PBL average CH4 should overestimate emissions. In fact, all the ingredients are available to quantify this link and assess the impact of transport biases on emissions. It raises the question why this is not done. Is it an important factor explaining the range of emission that are found or not?

Following the suggestion of the reviewer we analyzed the relationship between the estimated relative bias
(based on the enhancement compared to the background integrated over the boundary layer) and the model emissions in the area around the regular aircraft profiles sites. The analysis showed significant correlations between model emissions and estimated model bias. We will include this analysis in the revised version.

We have included now an analysis of the correlation between the derived relative bias and the regional model emissions around the aircraft sites (new Figure 16S in the supplementary material, and discussion at the end of section 4.2.

**SPECIFIC COMMENTS**

35

page 4, line 6: Which targets are set by the quality control mentioned here? Are they met?

No specific threshold values have been set. The typical range for the "working standard repeatability" is ~1-4 ppb. Since this "working standard repeatability" is used by the inverse models, measurements with higher "working standard repeatability" are weighted less in the inversion.

No change in the manuscript regarding this point.

**5**

page 5, line 16: Using constant a priori flux uncertainties also? How do these emissions / uncertainties relate to those of the other scenarios?

For inversion S3 very large uncertainties of the homogeneous a priori fluxes were assumed (ranging between 500% and 600% per grid-cell and month; see model description in the supplementary material) in order to give

10 the inversion enough degree of freedom to retrieve regional emission hot spots (which have much higher emissions than the applied homogeneous a priori fluxes). In contrast, the assumed uncertainties per grid cell and months are much smaller for the other scenarios (typically 100%).

We have added a short reference to the supplementary material, where the specific settings for inversion S3 are described in more details for the individual models.

15

page 5, line 24: Do the regional models (apart from NAME) prescribe boundary conditions, or allow further optimization?

Apart from NAME, the boundary conditions are further optimized also in CHIMERE, while the other regional models used prescribed boundary conditions. These boundary conditions were derived from optimized

20 concentrations of global inversion systems (STILT: from TM3, COMET: from TM5-4DVAR, CHIMERE: from LMDZ).

The information about the optimization of the boundary conditions been added in section 3.2.

page 8, line 10-15: It would be good to mention some typical numbers here for the bottom up and top down derived seasonal amplitudes (it is not so clear to see from figure 4)

25 We will add the numbers of the derived seasonal amplitudes in the revised version.

Instead of including the numbers, we have added now the new Figure 5S which visualizes the mean seasonal cycles for all scenarios.

**page 8, line 30-35: How about the seasonality in the energy sector? (domestic heating etc.)**

- 30 No or only small seasonal variations were found in the limited number of studies investigating natural gas distribution system [Wennberg et al., 2012; McKain et al., 2014]. Wong et al. [2016] argued that "the natural gas distribution pipeline system is pressure-regulated at several points, and leakage should be independent of consumption to first order", but that natural gas storage facilities may have seasonally varying leakage rates, depending on energy demands.
- 35 Reference to [McKain et al., 2015] has been added.

page 9, line 7: The difference between the observed vs simulated amplitude of variability (as used in Taylor diagrams for instance) provides a piece of information that is more independent from correlation as the RMS that is used here.

5 Following the suggestion of the reviewer we will analyze also the difference between the observed vs simulated amplitude of variability.

We have extended Figure 8S (previous Figure 6S), including an additional panel with the ratio between modelled and observed standard deviations.

**10 **1.3 Anonymous Referee #3**

**Summary/General comments:**

The manuscript presents 'top-down' optimized methane emissions for Europe for the 2006-2012 time period. A new, harmonized 18 site-monitoring network is used with seven inverse models and four experiments. Optimized emissions are reported (and are overall consistent between top-down and bottom-up), biases are assessed using

15 aircraft data, and the inference of a non-negligible wetland source is intimated. Overall it is interesting and important work to pursue. It is not easy to use this many different model/inverse approaches to one regional question, and this can potentially provide substantially more information and understanding for how to best quantify fluxes with atmospheric observations. This paper is well-placed in ACP.

We thank the reviewer for the very positive overall evaluation of our study.

20 However, there are a couple important gaps that need to be addressed before I can recommend publication. Most importantly, the description of different models and inverse methods is somewhat lacking, this should be a central element of this work, and this needs to be improved before I can recommend publication

The inverse modelling system are described in the supplementary material, summarizing the main elements of each system. Furthermore, all seven inverse models are described comprehensively in separate specific papers.

25 Nevertheless, we will include some further details in the description of the models.

Some further details (e.g., about the a priori probability density functions and assumed observations errors) were added / complemented in the supplementary materials.

**30 Major comments:**

Models/Inverse methods: There is limited discussion of the different models, and specifically, of the inverse methodology being employed by each model. I understand much of this is referenced to various previous publications, and the supplement does go through each model independently, but it is important for the reader to see more comparative details in this manuscript to be able to understand the differences between

35 models/inversions and possible nuanced causes. A succinct but clear description in its own section of the different inverse approaches used and the subtle "expert-user" choices made to define the inversion would be essential. For example the prior uncertainties and correlations lengths, which are defined differently in the different inversions, could be rather impactful on the results. How were these different priors chosen, and how important is this choice? The authors have conducted multiple experiments – they need to better convey to the reader the differences between the inversions and experiments so we can better assess the meaning of

5 similar/different results. In many ways this could be one of the biggest contributions of this paper.

The specific settings of the individual inverse models are indeed largely "expert-user" choices. For many models the sensitivity of derived emissions on these settings were investigated in more detail (and described in the papers of the individual inverse modelling systems). E.g. for TM5-4DVAR different spatial correlation lengths (between 100 and 300 km) were analyzed [Bergamaschi et al., 2010], showing an overall only very small impact

10 on the derived emissions. In the present study, the philosophy was to prescribe only the basic settings for the inversions, such as a priori emission inventories, observational data sets, and inversion time period.

The main objective of this study is to use the model ensemble to provide more realistic overall uncertainty estimates (from the range of the inverse models), rather than investigating the sensitivity of individual inversion results on specific settings of the individual models. Given the large fundamental differences of the different

15 inverse models (e.g. grid based inversion in TM5-4DVAR compared to optimization of larger pre-defined larger regions and different land-ecosystem types in the TM5-CTE (ensemble Kalman filter), it would not be possible to apply fully consistent settings in the different models.

The different inversions of this study investigate the impact of the different set of stations and the use of 'a priori' information. The different settings for the 4 inversion experiments are summarized in Table 2 and described in section 3.1

20 described in section 3.1.

We have included a reference [Bergamaschi et al., 2010] for the sensitivity of derived CH4 emissions on spatial correlations (and OH fields) for TM5-4DVAR (section 1.1 supplementary material).

Sensitivity of network to domain: Western Europe has the highest density of observation sites, and measurement density (and sensitivity to emissions) falls off rapidly in other regions of Europe. Given this, how appropriate is it to lump the entirety of the domain together? I'd like to see a little more discussion of the sensitivity of the network and therefore dependence of prior/assumptions in some of the domains. Another way to consider this question is how many regions can the network distinguish, and how do these regions compare with geopolitical domains? This impacts my next point.

- 30 Indeed the available stations are not evenly distributed across Europe, and the observational coverage is relatively sparse in southern Europe and Scandinavia. The fact that inversion S3 yields similar estimates for the emissions of Northern and Southern Europe (for most models; however lower estimates for NAME) compared to the other inversions (which include the detailed emission inventories as a priori) suggests that nevertheless the limited observations provide also some constraints on the total emissions from these sub-regions. We did
- 35 not perform specific sensitivity experiments in this study, but we will include some more discussion of the network coverage (and the limited observational constraints in southern Europe) in the revised version.

We have included a short discussion of the network coverage in section 4.1 ("Despite the significantly larger number of European monitoring stations in the present study, however, we emphasize that the available stations do not very well cover the whole EU-28 area. Consequently, the emissions especially from Southern Europe remain poorly constrained ")

40 Europe remain poorly constrained.").

Importance of wetlands: I'm not sure if from this analysis alone the authors can conclude substantial wetland source are or are not required to match observations. The largest prior wetland estimate (and seasonality) is in Northern Europe, where there are few observation points and the inverted seasonality is actually smaller than WETCHIMP models. When aggregating all of Europe together, it would appear the added emissions and

5 seasonality from WETCHIMP is helpful in bringing bottom-up and top-down closer together – but given this point of spatial/seasonal errors in the Northern Europe domain I'm not sure this overall improvement is indicative of a better representation or coincidence where the inversion finds large seasonality in other regions of Europe where WETCHIMP models do not expect significant wetland sources. I would think the authors should tone done the statement of wetlands importance in the abstract, and also would like to see further defense of the 10 seasonality signal observed and attribution that it must be wetlands.

Indeed the spatial distribution of wetlands in Europe in the WETCHIMP ensemble is not fully consistent with the results from the inverse models and most inverse models (except TM5-CTE) show a smaller amplitude of the seasonal variations than the mean of the WETCHIMP ensemble. Nevertheless, the WETCHIMP ensemble estimates significant wetland emissions also in western / southern / eastern Europe (2.5 (0.4-5.1) Tg CH4 yr-1;

- 15 see also our reply to reviewer #1) and the seasonal cycles derived by 4 models (TM5-4DVAR, TM5-CTE, TM3-STILT, and LMDZ) are broadly consistent with the range of seasonal variations of the WETCHIMP ensemble (although indeed the amplitude of the mean seasonal cycles of WETCHIMP are smaller for western / southern / eastern Europe). We fully agree that the uncertainties of wetland emissions remain very high (as directly evident from the very different spatial distributions of the individual WETCHIMP inventories (see Figure 4S).
- 20 This has been mentioned in the text, but will be further emphasized in the revised version.

Also inversion S3 (which was performed without using detailed bottom-up inventories as 'a priori'), shows significant seasonal cycles in derived emissions (for EU-28 and all European subregions (but relatively small in southern Europe)), which confirms that the derived seasonal cycles are driven by the observations (and not by the a priori emissions).

25 We also agree that uncertainties remain in the attribution of the seasonal cycle to wetlands, since some anthropogenic sources may also exhibit some (smaller) seasonal variations (see also our reply to reviewer #1). We will emphasize the caveats of our wetland hypothesis more clearly in the revised version (including the abstract).

We have included a new figure (Figure 5S) with the mean seasonal cycles derived in the different inversions
(including S3). Furthermore, we emphasize now more clearly the caveats of our wetland hypothesis in the abstract ("However, the contribution of natural sources remains rather uncertain, especially their regional distribution.") and the conclusions ("Furthermore, it needs to be emphasized that wetland inventories have larger uncertainties and show large differences in the spatial distribution of CH4 emissions.").

**35 **2.** Further updates of the manuscript**

We have further refined our method to evaluate the model biases (section 4.2):

(1) In order to reduce the impact of potential errors of the background on the calculation of the relative bias (section 4.2) we have increased the threshold of  $\Delta c_{OBS, BL}$  and  $\Delta c_{OBS, COL}$  from 10 ppb to 20 ppb.

(2) For the evaluation of the enhancements of the measurements vs. background we use for NAME and TM3-STILT now the background evaluated by TM5-4DVAR for the NAME and TM3-STILT domains. For the evaluation

of the simulated CH4 enhancements of NAME and TM3-STILT, we use now the actual background used in NAME and TM3-STILT (instead of the TM5-4DVAR background used previously). The relative biases derived with these updates for NAME and TM3-STILT are considered much more accurate, and reveal a significant negative regional bias for these two models. The NAME and TM3-STILT backgrounds are evaluated and discussed now in

5 some more detail in the paper (new Figures 14S and 15S).

Furthermore, some updates of the LMDZ inversions were included in the revised version:

- for LMDZ inversion S1 uncertainty estimates have been included.

- LMDZ inversion S3 has been updated (use a priori uncertainty of 600% (instead of 200% used previously)). As a result also mean total CH4 emissions for the EU-28 have slightly changed (from 26.7 to 26.8 Tg CH4 yr-1)

10 In addition, we updated the NOAA AGGI and recent atmospheric CH4 in the introduction.

**Updates of Figures:**

Figure 3: Include uncertainty estimates for LMDZ inversion S1. Update LMDZ inversion S3.

Figure 4: Include uncertainty estimates for LMDZ inversion S1.

Figure 5: Include also background CH4 used in NAME (based on MHD data) and TM3-STILT (based on TM3)

15 **Figure 6:** Update evaluation of relative bias using threshold of  $\Delta c_{OBS, BL}$  of 20 ppb (instead of 10 ppb). Updated evaluation of model enhancements / relative bias for NAME and TM3-STILT. Linear fits have been removed (as they were not discussed in the paper).

**Figure 7:** Update evaluation of relative bias using threshold of  $\Delta c_{OBS, BL}$  and  $\Delta c_{OBS, COL}$  of 20 ppb (instead of 10 ppb). Updated evaluation of model enhancements / relative bias for NAME and TM3-STILT.

**20 Supplementary material**

Figure 3S: update S3 inversion LMDZ (use a priori uncertainty of 600% (instead of 200% used previously)).

Figure 5S: new figure with mean seasonal cycles for all scenarios (as suggested by reviewer 1 and 2)

Figure 6S (previous Figure 5S): include uncertainty estimates for LMDZ S4

Figure 7S: new figure with analysis of trends (as suggested by reviewer 2)

25 **Figure 8S (previous Figure 6S):** include also analysis of ratio of model standard deviation to observed standard deviation (as suggested by reviewer 2)

Figure 9-12S (previous Figure 7-10S): Update evaluation of enhancements vs background using thresholds of  $\Delta c_{OBS, BL} = 20$  ppb and  $\Delta c_{OBS, COL} = 20$  ppb (instead of 10 ppb). The integrated enhancements of the measurements vs background (evaluated by TM5-4DVAR) are evaluated now separately for the TM5 zoom domain and the

30 NAME and STILT model domains. Furthermore, for the NAME and TM3-STILT model, the integrated model enhancement is now evaluated using the NAME background (based on MHD baseline data) and TM3 background, respectively. **Figure 13S (previous Figure 11S):** Update evaluation of relative bias using threshold of  $\Delta c_{OBS, COL}$  of 20 ppb (instead of 10 ppb). Updated evaluation of model enhancements / relative bias for NAME and TM3-STILT. Linear fits have been removed (as they were not discussed in the paper).

New Figure 14S: Evaluation of CH4 background for TM5-4DVAR, NAME, and TM3-STILT, comparing model

5 simulations with the aircraft observations for events with very low simulated contribution (≤ 3 ppb) from European CH4 emissions.

New Figure 15S: Background CH4 at European monitoring stations for TM5-4DVAR, NAME, and TM3-STILT.

**New Figure 16S:** Correlation between relative bias and regional model emissions around the aircraft sites (as suggested by reviewer #2).

**10**

[revised manuscript text omitted]
 n-small average positive-relative bias (rbBL between -7% and 10%) at the three aircraft sites(TM5-4DVAR: 20%; TM5-CTE: 22%; LMDZ: 30%), indicating that these models likely overestimate the regional emissions. In contrast, TM3-STILT and NAME have significant negative relative biases (TM3-STILT: rbBL between -13% and -24% for the three sites; NAME rbBL = -30% for ORL and HNG).very small biases of only 1% and 3%, respectively, at this profile site. Also at HNG, three models show some positive bias on average (TM5-4DVAR: 14%, LMDZ: 14%, LM
- 10 16%, TM3-STILT: 11%), while the bias of NAME and TM5-CTE is close to zero. In contrast, at BIK all models show small relative bias (TM5-4DVAR: -2%; TM3-STILT: 5%; TM5-CTE: 6%; LMDZ: 6%; NAME: not available). These negative biases are likely related to the positive bias in the background CH4 used for NAME and TM3-STILT (see above), since the regional models invert the difference between the observations and the assumed background. In fact, also at most continental atmospheric monitoring stations, the background used for NAME and TM3-STILT is significantly higher
- 15 (~10 ppb) compared to the TM5-4DVAR background (Figure 15S). The 'relative bias' is also extracted separately for different seasons (right panel of Figure 6). At ORL, all models have relative biases close to zero in spring (March May), while larger relative biases are visible during other seasons. Apart from this feature, tThere is no clear seasonal cycle in the relative bias apparent and the variability between the different seasons is generally small-at HNG and BIK (data points at BIK for DJF are considered not significant as they are from one single profile
- 20 only). From this analysis there is no evidence that the seasonal cycle of emissions derived by four inverse models (TM5-4DVAR, TM5-CTE, TM3-STILT, and LMDZ; see section 4.1) with clear maxima in summer could be due to a seasonal bias in the transport models. At the same time, however, NAME, which calculates much smaller seasonal variations of emissions, also shows no only small seasonal variations of the average bias at ORL and - At HNG, NAME has - 20 40% lower average bias between December and May compared to June to November, which seems to contrast with the seasonality of the emissions
- 25 derived by NAME; however\_However, especially at this siteHNG the total number of profiles is rather small (n=22), which limits the analysis of potential seasonal transport biases.

Figure 118-138 shows the relative bias of the CH4 enhancements integrated over the lower troposphere, defined as:

[revised manuscript text omitted]

| ID   | station name     | data provider            | lat   | lon         | alt  | s. h.   | ST     | S1 | S2 | S3 | S4 |
|------|------------------|--------------------------|-------|-------------|------|---------|--------|----|----|----|----|
| ZEP  | Ny-Alesund       | InGOS/NILU 1  | 78.91 | 11.88       | 474  | 15      | Ι      | •  | •  | •  | •  |
|      |                  | NOAA                     | 78.91 | 11.88       | 474  | 5       | D      | •  | •  | ٠  |    |
| SUM  | Summit           | NOAA                     | 72.60 | -38.42      | 3210 | 5       | D      | •  | •  | •  |    |
| PAL  | Pallas           | InGOS/FMI 2   | 67.97 | 24.12       | 565  | 7       | Ι      | •  | •  | •  | •  |
|      |                  | NOAA                     | 67.97 | 24.12       | 560  | 5       | D      | •  | •  | ٠  |    |
| ICE  | Storhofdi,       | NOAA                     | 63.40 | -20.29      | 118  | 9       | D      | •  | •  | ٠  |    |
| VKV  | Voeikovo         | InGOS/MGO 3   | 59.95 | 30.70       | 70   | 6       | Ι      |    | ٠  | ٠  | ٠  |
| TTA  | Angus            | InGOS/UoE 4   | 56.55 | -2.98       | 313  | 222     | Ι      | •  | •  | ٠  | •  |
| BAL  | Baltic Sea       | NOAA                     | 55.35 | 17.22       | 3    | 25      | D      |    |    |    |    |
| LUT  | Lutjewad         | InGOS/CIO 5   | 53.40 | 6.35        | 1    | 60      | Ι      | •  | •  | •  | •  |
| MHD  | Mace Head        | InGOS/UoB 6   | 53.33 | -9.90       | 25   | 15      | Ι      | •  | •  | •  | •  |
|      |                  | NOAA                     | 53.33 | -9.90       | 5    | 21      | D      | •  | •  | ٠  |    |
| BIK1 | Bialystok        | InGOS/MPI 7   | 53.23 | 23.03       | 183  | 5       | Ι      |    |    |    |    |
| BIK2 |                  |                          |       |             |      | 30      | I      |    |    |    |    |
| BIK3 |                  |                          |       |             |      | 90      | I      |    |    |    |    |
| BIK4 |                  |                          |       |             |      | 180     | Ι      |    |    |    |    |
| BIK5 |                  |                          |       |             |      | 300     | Ι      | •  | •  | •  | •  |
| CBW1 | Cabauw           | InGOS/ECN 8   | 51.97 | 4.93        | -1   | 20      | Ι      |    |    |    |    |
| CBW2 |                  |                          |       |             |      | 60      | Ι      |    |    |    |    |
| CBW3 |                  |                          |       |             |      | 120     | Ι      |    |    |    |    |
| CBW4 |                  |                          |       |             |      | 200     | Ι      | •  | •  | •  | •  |
| OXK1 | Ochsenkopf       | InGOS/MPI 7   | 50.03 | 11.82       | 1022 | 23      | Ι      |    |    |    |    |
| OXK2 |                  |                          |       |             |      | 90      | I      |    |    |    |    |
| OXK3 |                  |                          |       |             |      | 163     | I      | •  | •  | ٠  | ٠  |
| OXK  |                  | NOAA                     | 50.03 | 11.82       | 1022 | 163     | D      |    | -  |    |    |
| HEI  | Heidelberg       | InGOS/IUP 9   | 49.42 | 8.67        | 116  | 30      | I      | •  | •  | •  | •  |
| KAS  | Kasprowy Wierch  | InGOS/AGH 10  | 49.23 | 19.98       | 1987 | 2       | 1      |    | •  | •  | •  |
| LPO  | Ile Grande       | RAMCES                   | 48.80 | -3.58       | 20   | 10      | D      | •  | •  | •  |    |
| GIF  | Gif sur Yvette   | InGOS/LSCE 11 | 48.71 | 2.15        | 160  | 7       | Ι      | •  | •  | ٠  | ٠  |
| TRN1 | Trainou          | InGOS/LSCE 11 | 47.96 | 2.11        | 131  | 5       | I      |    |    |    |    |
| TRN2 |                  |                          |       |             |      | 50      | I      |    |    |    |    |
| TRN3 |                  |                          |       |             |      | 100     | I      |    |    |    |    |
| TRN4 |                  | L COC/TED + 12           | 45.01 | 5 01 | 1005 | 180     | I      |    | •  | •  | •  |
| SCH  | Schauinsland     | InGOS/UBA 12  | 47.91 | 7.91        | 1205 | 8       | l
D | •  | •  | •  | •  |
| НРВ  | Hohenpeissenberg | NOAA                     | 47.80 | 11.01       | 985  | 5       | D      | •  | •  | •  |    |
| HUN  | Hegyhatsal       | InGOS/HMS 13  | 46.95 | 16.65       | 248  | 96      |        | •  | •  | •  | •  |
| HUN  |                  | NOAA                     | 46.95 | 16.65       | 248  | 96
7 | D      | •  | •  | •  |    |
| JFJ  | Jungfraujoch     | InGOS/EMPA 14 | 46.55 | 7.98        | 3575 | 5       | I      | •  | •  | •  | •  |
| IPR  | Ispra            | InGOS/JRC 13  | 45.81 | 8.63        | 223  | 15      | I      |    | •  | •  | •  |
| PUY  | Puy de Dome      | InGOS/LSCE 11 | 45.77 | 2.97        | 1465 | 10      | l
D |    | •  | •  | •  |
| PUY  |                  | RAMCES                   | 45.77 | 2.97        | 1465 | 10      | D      | •  | •  | •  |    |
| BSC  | Black Sea        | NOAA                     | 44.17 | 28.68       | 0    | 5       |        |    |    |    |    |
| PDM  | Pic du Midi      | RAMCES                   | 42.94 | 0.14        | 2877 | 10      | D      | •  | •  | •  |    |
| BGU  | Begur            | RAMCES                   | 41.97 | 3.23        | 13   | 2       | D      | •  | •  | •  |    |
| LMP  | Lampedusa        | NOAA                     | 35.52 | 12.62       | 45   | 5       | D      | •  | •  | •  |    |
| FIK  | Finokalia        | RAMCES                   | 35.34 | 25.67       | 150  | 15      | D      | 1  | •  | •  |    |

1 Norwegian Institute for Air Research, Norway
 2 Finnish Meteorological Institute, Helsinki, Finland

- 3 Main Geophysical Observatory, St. Petersburg, Russia
- 4 University of Edinburgh, Edinburgh, UK
- 5 Center for Isotope Research, Groningen, Netherlands
- 6 University of Bristol, Bristol, UK
- 7 Max Planck Institute for Biogeochemistry, Jena, Germany
   8 Energy research Centre of the Netherlands, Petten, Netherlands
   9 Institut f
  ür Umweltphysik, Heidelberg, Germany
  - 10 University of Science and Technology, Krakow, Poland
  - 11 Laboratoire des Sciences du Climat et de l' Environnement, Gif-sur-Yvette, France
  - 12 Umweltbundesamt Germany, Messstelle Schauinsland, Kirchzarten, Germany
    - 13 Hungarian Meteorological Service, Budapest, Hungary
    - 14 Swiss Federal Laboratories for Materials Science and Technology, Dübendorf, Switzerland
    - 15 European Commission Joint Research Centre, Ispra, Italy

15

10

20

**Table 2: CH4 inversions**

| $\mathbf{a}$ | ~        |
|--------------|----------|
|              |   |
| -            | -        |

| inversion | a priori emissions                                        | period    | InGOS station | NOAA+RAMCES          |  |  |
|-----------|-----------------------------------------------------------|-----------|---------------|----------------------|--|--|
|           |                                                           |           |               | discrete air samples |  |  |
| S1        | EDGARv4.2FT-InGOS                                         | 2006-2012 | base          | •                    |  |  |
| S2        | EDGARv4.2FT-InGOS                                         | 2010-2012 | extended      | •                    |  |  |
| S3        | no detailed a priori inventory 1 | 2010-2012 | extended      | •                    |  |  |
| S4        | EDGARv4.2FT-InGOS                                         | 2010-2012 | extended      | -                    |  |  |

**$^{1}$ see section 3.1**

**30 **Table 3:** Atmospheric models**

| Model     | Institution | Resolution of transport model:                     | Vertical        | Meteorology                        | Background CH 4    |
|-----------|-------------|----------------------------------------------------|-----------------|------------------------------------|-------------------------------|
|           |             | Horizontal (lon $\times$ lat)                      |                 |                                    | (regional models)             |
| TM5-4DVAR | EC JRC      | Europe: $1^{\circ} \times 1^{\circ}$               | 25              | ECMWF ERA-INTERIM                  |                               |
|           |             | Global: $6^{\circ} \times 4^{\circ}$               |                 |                                    |                               |
| TM5-CTE   | FMI         | Europe: $1^{\circ} \times 1^{\circ}$               | 25              | ECMWF ERA-INTERIM                  |                               |
|           |             | Global: $6^{\circ} \times 4^{\circ}$               |                 |                                    |                               |
| TM3-STILT | MPI-BGC     | Europe: $0.25^{\circ} \times 0.25^{\circ}$ (STILT) | 61 (STILT)      | ECMWF operational analysis (STILT) | TM35        |
|           |             | Global: $5^{\circ} \times 4^{\circ}$ (TM3)         | 26 (TM3)        | ECMWF ERA-INTERIM (TM3)            |                               |
| LMDZ      | LSCE        | Europe: $\sim 1.2^{\circ} \times 0.8^{\circ}$      | 19              | Nudged to ECMWF ERA-INTERIM        |                               |
|           |             | Global: $\sim 7^{\circ} \times 3.6^{\circ}$        |                 | -                                  |                               |
| NAME      | Met Office  | 0.5625° × 0.375° 1                      | 313             | Met Office Unified Model (UM)      | based on measurements at      |
|           |             | 0.3516° × 0.2344° 2                     | 59 4 |                                    | Mace Head 6        |
| CHIMERE   | LSCE        | $0.5^{\circ} \times 0.5^{\circ}$                   | 29              | ECMWF ERA-INTERIM                  | $\underline{\text{LMDZ}^{6}}$ |
| COMET     | ECN         | $0.17^{\circ} \times 0.17^{\circ}$                 | 60              | ECMWF ERA-INTERIM                  | TM5-4DVAR                     |

1 for simulation period 01/2006-03/2010

- 2 for simulation period 03/2010-12/2012
- 3 for simulation period 01/2006-10/2009
- 4 for simulation period 10/2009-12/2012

5 coupling based on the method of *Rödenbeck et al.* [2009],

40

35

<sup>6 further optimized in the inversion

---

## Referee Report (RR1)

Re-review of "*Inverse modelling of European CH$_4$ emissions during 2006–2012 using different inverse models and reassessed atmospheric observations*" by Bergamaschi *et al.*

I do not find the changes between this version and the original manuscript to be compelling enough to change my original stance on this manuscript. It is not clear to this reviewer that this manuscript contributes much to the current literature.

Here were my three concerns from the original manuscript:

1. I find the wetland hypothesis wholly unconvincing

2. the methods description is poor, making it hard to gain any insight from the different inversions

3. it's not clear to this reviewer that their "novel" approach to estimate bias is actually an advancement

**Point #1:**
The authors have added one sentence to the abstract and one sentence to the conclusions.

**Point #2:**
The methods description is still poor. There is still just a single paragraph in the main text describing the inversions. It is left to the reader to guess at why the inverse models obtain, in some cases, radically different emissions. Figure 2 is a good example of this. The emissions from COMET look totally different from the others (e.g., why is there a source in Northern Poland that isn't in the prior or any other model?). Figure 2 seems to only be mentioned a single time in the manuscript (in the first paragraph of Section 4.1).

In the author's response they state: *"But this [understanding the differences between top-down emissions] is actually not the goal of this study (and would require further specific modelling experiments). The objective of this study is to use the model ensemble to provide more realistic overall uncertainty estimates (from the range of the inverse models) and to evaluate the model performance by validation against independent observations."* Was this not already done in the 2015 paper by many of the same authors (*Bergamaschi et al., "Top-down estimates of European CH$_4$ and N$_2$O emissions based on four different inverse models", ACP, 2017*)? The authors have just added in a few more models and a few more years of data.

In general, the conclusions drawn do not seem to be in-line with the analysis. For example, the conclusions of this manuscript state (final paragraph): *(2) transport models need to be further improved, including their spatial resolution and in particular the simulation of vertical mixing*, aren't some of these models finer-resolution then others and include different treatments of vertical mixing? Do these models actually perform better?

**Point #3:**
Regarding the 3rd point, their "novel" approach to estimate bias is not particularly useful for estimating biases (as the authors claim). The difference between simulated and measured enhancements is the term that defines the model-data mismatch in the cost-function. As I showed in my previous review, $c_{\mathrm{obs}} - c_{\mathrm{mod}} \equiv \Delta c_{\mathrm{obs}} - \Delta c_{\mathrm{mod}}$. Following this, if there were no difference between the simulated and measured enhancement then the inverse model would not deviate from the prior. Stated another way, if $\Delta c_{\mathrm{obs}} = \Delta c_{\mathrm{mod}}$ then the top-down emissions would be equal to the prior.

---

## Author Response (AR2)

**Response to re-review of Anonymous Referee #1 and Co-Editor Decision letter for ACP manuscript No.: acp-2017-273 [Peter Bergamaschi et al., Inverse modelling of European CH$_4$ emissions during 2006-2012 using different inverse models and reassessed atmospheric observations]**

5  This document describes the detailed point-by-point response to re-review of Anonymous Referee #1 and Co-Editor Decision letter by Jens-Uwe Grooß (13 Nov 2017)

(1) *comments from Referee / Co-Editor [in italics]*

(2) **author's response [bold]**

(3) **author's changes in manuscript [bold blue].**

**1. Response to re-review of Anonymous Referee #1**

*I do not find the changes between this version and the original manuscript to be compelling enough to change my*
15  *original stance on this manuscript. It is not clear to this reviewer that this manuscript contributes much to the current literature.*

**We do not agree with this statement, since we had provided a point-by-point reply addressing all points raised by all 3 referees and revised significantly the manuscript.**

20  **We believe that our paper makes a significant contribution to the literature in the field of inverse modelling of European / regional CH$_4$ emissions and includes many new elements compared to the previous [*Bergamaschi et al., ACP,* 2015] paper:**

**(1) use of the new quality-controlled and harmonized InGOS CH$_4$ in-situ data set including 18 stations. To our knowledge this is the largest (and most consistent) data set of European atmospheric measurements used**
25  **in any study of top-down estimates of European CH$_4$ emissions so far.**

**(2) extension of time series (now 7-years period 2006-2012, compared to 2-years period in [*Bergamaschi et al.*, 2015], including a short analysis of CH$_4$ trends.**

**(3) detailed analysis of potential contribution of natural sources**

**(4) comprehensive validation of model results using an extended set of aircraft observations, providing for**
30  **the first time quantitative estimates of potential biases in derived regional emissions.**

*Point #1 ("I find the wetland hypothesis wholly unconvincing"): The authors have added one sentence to the abstract and one sentence to the conclusions.*

**We did not only include the additional sentences in the abstract and the conclusions, which clearly emphasize the uncertainties of the wetland emissions, especially regarding their spatio-temporal emission distribution (furthermore these uncertainties are discussed in sections 4.1 (already in the original discussion paper)). Following the suggestion of this referee we had included an analysis of the seasonality of derived CH$_4$**

5 **emissions in scenario S3 (which is not using any detailed a priori inventory nor any a priori seasonal cycle; see section 4.1 and new Figure 5S, which confirms that the derived emissions are driven by the observations and not by the a priori inventories). Furthermore, we had addressed the further specific point raised by this referee on this topic, especially the potential impact of the background (see section 4.2, new Figure 14S).**

10 *Point #2 ("the methods description is poor, making it hard to gain any insight from the different inversions"): The methods description is still poor. There is still just a single paragraph in the main text describing the inversions.*

**As explained in our response to the reviewers, the inverse modelling system are described in the supplementary material (SM), section 1 "Atmospheric models" (summarizing the main elements of each**

15 **system; this description extends over more than 4.5 pages). Furthermore, all seven inverse models are described comprehensively in separate specific papers (see references in the SM).**
**Since for most models used in this study only smaller updates were applied (compared to previously published applications), we think that is more appropriate to keep the summarizing description of the individual models in the SM.**

20 **However, following the request of this referee we had updated the description in the SM (e.g. the applied a priori probability density functions (pdf's) for the individual models, and the assumed uncertainties for the observations (including estimates of model errors). Furthermore, we had included the applied boundary conditions (background) in Table 3 and the information about the optimization of the background in the revised main paper.**

25 **Following also the request of the co-editor, we included further model details in the main paper (section 3.2 / table 3), as detailed below (see under "2. Response to Co-Editor Decision letter by Jens-Uwe Grooß (13 Nov 2017)")**

*It is left to the reader to guess at why the inverse models obtain, in some cases, radically different emissions. Figure*

30 *2 is a good example of this. The emissions from COMET look totally different from the others (e.g., why is there a source in Northern Poland that isn't in the prior or any other model?). Figure 2 seems to only be mentioned a single time in the manuscript (in the first paragraph of Section 4.1).*

**A significant part of the 'visual' differences in the spatial patterns between different models is related to**

35 **different spatial resolutions (and, as explained in the text, for NAME related to the averaging of emissions at larger distances from the observations). Integrated over larger areas (e.g. whole EU-28), the models show a remarkable consistency (apart from the generally lower CH$_4$ emissions derived by NAME). Differences on smaller spatial scales are probably partly due to differences in model transport and different weighting of**

the observations (i.e. different assumptions of model-data mismatch errors), but may reflect to some extent some noise of the inverse modelling systems.

**In order to address this specific new comment of the reviewer, we have added the following short additional paragraph in the manuscript (in section 4.1):**

5    **"Apart from this specific feature of the NAME model, also some further differences in the spatial patterns derived by the different models are apparent. One example are the relatively high emissions derived by the COMET model in north-western Poland / north-eastern Germany. Such differences on smaller spatial scales are probably partly due to differences in model transport and different weighting of the observations (i.e. different assumptions of model-data mismatch errors), but may reflect to some extent also some noise of the**

10    **inverse modelling systems."**

*In the author's response they state: "But this [understanding the differences between top-down emissions] is actually not the goal of this study (and would require further specific modelling experiments). The objective of this study is to use the model ensemble to provide more realistic overall uncertainty estimates (from the range of the*

15    *inverse models) and to evaluate the model performance by validation against independent observations." Was this not already done in the 2015 paper by many of the same authors (Bergamaschi et al., "Top-down estimates of European CH4 and N2O emissions based on four different inverse models", ACP, 2017)? The authors have just added in a few more models and a few more years of data.*

20    **We assume that the referee refers here to our [*Bergamaschi et al.*, ACP, 2015] (and not 2017) paper. As outlined above we think that the current paper provides many new elements and provides a significantly extended and improved analysis of European CH$_4$ emissions. What the referee calls "just added... a few more years" is a significant extension of the time period (covering now the 7-years period 2006-2012, while in [*Bergamaschi et al.*, 2015] only the 2 years 2006 and 2007 were analyzed).**

*In general, the conclusions drawn do not seem to be in-line with the analysis. For example, the conclusions of this manuscript state (final paragraph): (2) transport models need to be further improved, including their spatial resolution and in particular the simulation of vertical mixing, aren't some of these models finer-resolution then others and include different treatments of vertical mixing? Do these models actually perform better?*

**We had presented detailed conclusions in-line with our analysis in section "5. Conclusions". The referee refers here to our more general conclusions at the end (which are rather recommendations to further improve top-down estimates in the future). In our study the models with higher spatial resolution do not perform better than TM5-4DVAR with resolution of 1x1 degrees (see short discussion at beginning of section**

35    **4.2). Nevertheless we consider the further development of high-resolution models essential to further improve the top-down estimates (e.g. [*Henne et al.,* 2016]).**

*Point #3 ("it's not clear to this reviewer that their "novel" approach to estimate bias is actually an advancement"): Regarding the 3rd point, their "novel" approach to estimate bias is not particularly useful for estimating biases (as the authors claim). The difference between simulated and measured enhancements is the term that defines the model-data mismatch in the cost-function. As I showed in my previous review, $c_{obs} - c_{mod} \equiv \Delta c_{obs} - \Delta c_{mod}$. Following this, if there were no difference between the simulated and measured enhancement then the inverse model would not deviate from the prior. Stated another way, if $\Delta c_{obs} = \Delta c_{mod}$ then the top-down emissions would be equal to the prior.*

**As explained in our response to the reviewers we do not agree with the statement of the referee. The term mentioned by the reviewer is indeed part of the cost function, which the inverse modelling systems aim to minimize. However, this applies to the observations that are actually used (assimilated) in the inversions, while we are analyzing here independent observations that were not used in the inversion - which is a common method to validate inverse models. The concept of using independent observations for validation of inverse models is described in detail in the recent review paper by *Michalak et al.,* [2017] and has been applied in many studies (e.g. [*Alexe et al.*, 2015; *Monteil et al.*, 2013; *Houweling et al.*, 2014; *Bergamaschi et al.,* 2013]). The rationale behind this approach is to analyze, how well the inverse model perform in areas which are less constrained by the observations. As mentioned in our initial reply to the reviewer, and explained in section 4.2 of the manuscript, especially the validation of the vertical profiles (against independent aircraft profiles) is very important, since the inverse models assimilate only surface observation. Therefore, potential errors in the vertical mixing of the models can introduce significant biases in the derived emission.**

**Commonly, however, such comparisons against independent observations are performed to diagnose only qualitatively, if the inverse models have biases. The novel aspect of our method is that we provide for the first time quantitative estimates of the derived regional emissions. We had included in the revised version also an analysis of the correlation between the derived relative bias ($rb_{BL}$) and regional model emissions around the aircraft profile sites, which confirms that $rb_{BL}$ can be used to diagnose biases in the regional model emissions (new Figure 16S).**

**Since we consider Point #3 as absolutely invalid, no changes have been made in the revised version regarding this point.**

**2. Response to Co-Editor Decision letter by Jens-Uwe Grooß (13 Nov 2017)**

*1. It is not clear to me, why one model that seems to be always at the low end of the deduced CH4 emissions: the Lagrangian NAME model. Is that because of the Lagrangian formulation of the model? Or because the lack of observation stations?*

**The low emissions of the NAME model are likely - at least partly - related to the positive bias in the background $CH_4$ used for NAME (as discussed in section 4.2 and shown in the figures 14S (for the aircraft data), since the regional models invert the difference between the observations and the assumed background.**

**In fact, also at most continental atmospheric monitoring stations (which are applied in the inversion), the background used for NAME (and TM3-STILT) is significantly higher (~10 ppb) compared to the TM5-4DVAR background (Figure 15S).**

**The potential impact of the significant positive bias of the background in NAME (and TM3-STILT) on the derived $CH_4$ emissions is also mentioned in the conclusions of the revised version.**

**It would be certainly useful to perform additional tests with the NAME model using the TM5 baselines in order to quantify the impact of the baseline on the emissions. Unfortunately, however, the NAME team was not yet able to perform this additional test, since they had no further resources after the end of the InGOS project.**

**We don't see any obvious reason why the Lagrangian models should yield lower emissions compared to Eulerian models (however, to our knowledge, this has not been investigated in any study in a systematic way). In fact, also STILT is a Lagrangian model and yields similar emissions as the global / regional Eulerian models.**

**Regarding the observations: All models use the same observational data set (however, the details, how the observations are used in the inversions, differ, in particular the assumed model-data mismatch error, and hence the weighting of the individual observations in the different models).**

**Short general description of the applied the assumed model-data mismatch errors has been included in section 3.2 (see below)**

*2. Can you understand that the annual cycle of wetland emissions is reduced in Northern Europe with respect to the WETCHIMP study? and opposite than the annual cycle increases in the three other parts? Would you say that the WETCHIMP study is incorrect in that respect?*

**We can only speculate about the potential reasons. E.g. the $CH_4$ emissions in the wetland models are highly sensitive to the assumed assumed temperature dependence ($Q_{10}$ values), but also on water table and soil properties (in particular, soil organic carbon content). We note that for Northern Europe the seasonality in the a posteriori emissions derived by TM5-CTE are actually very similar compared to the WETCHIMP mean / median. Nevertheless, it is obvious that the other models derive smaller seasonal cycles in Northern Europe. For the other regions in Europe the derived seasonality is still broadly within the minimum-maximum range of WETCHIMP (even though the seasonality in the mean/median of the WETCHIMP is clearly smaller).**

**We would not say that WETCHIMP is 'incorrect', but clearly the uncertainties are very large, as reflected in the very different spatio-temporal emission patterns of the different individual WETCHIMP inventories.**

*To Point #2 of the review:*

*Although the model descriptions are updated in the supplement, I would likely ask you to add some more information to section 3.2. /table 2, such that in the paper the model diversity would be better understandable*

*without reading the supplement. That should be a few more sentences about the methods (Lagrangian/Eulerian) underlying data (if different) resolution etc.*

**We included further details in section 3.2 / table 3 of the main paper**

**- short description of the applied meteorological fields in the text**

**- short description of the applied inverse modelling technique (added both in the text and table 3)**

**- type of model (Eulerian / Lagragian) has now been added also in table 3 (was already in the text)**

**- short general description of the applied uncertainties of the observations (including the measurement and model uncertainties)**

**(model resolutions were already described in the text and included in table 3)**

*technical corrections:*

*line 11 4.3 (2.3-8.2) Tg CH4 yr -1*

**'Tg' has been added**

*line 26 globally averaged tropospheric CH4 mole fraction*

**'tropospheric' has been added**

**(although this is not exactly the average over the entire troposphere - but with the explanation following in the text ('global average from marine surface sites') it should be clear.)**

All models used the same observational data set described in section 2 (except the stations ZEP and ICE, that are outside the domain of some regional models and except the mountain stations JFJ, PDM and KAS, which were not used in the NAME inversions). For the stations with in-situ measurements in the boundary layer, most models assimilated only measurements in the early afternoon (between 12:00 and 15:00 LT), and for mountain stations only night-time measurements (between 00:00 and 03:00 LT) [*Bergamaschi et al.*, 2015]. However, NAME and COMET used observations at all times. The different models have different approaches to estimate the uncertainties of the observations (including the measurement and model uncertainties), which determine the weighting of the individual observations in the inversions. In general, the estimated model uncertainties depend on the type of station, and for some models (TM5-4DVAR and NAME) also on the specific synoptic situation. The individual inverse models modelling system are described in more detail in the supplementary material (section 1).

**4 Results and discussion**

**4.1 European CH$_4$ emissions**

Figure 2 shows the maps of the European CH$_4$ emissions (average 2010-2012) derived from the seven inverse models for inversion S4. The corresponding maps for inversions S1-S3 (available from five models) are shown in the supplementary material (Figures 1S-3S). In S1, S2, and S4, which are guided by the 'a priori' information from the emission inventories, the 'a posteriori' spatial distributions are usually close to the prior patterns on smaller scales (determined by the chosen spatial correlation scale lengths). The NAME inversion groups together grid cells for which the observational constraints are weak, i.e., it averages over increasingly larger areas at larger distances from the observations. Consequently, in the NAME inversion the 'fine structure' of the 'a priori' inventories disappears in areas which are not well constrained (e.g., Spain). Apart from this specific feature of the NAME model, also some further differences in the spatial patterns derived by the different models are apparent. One example are the relatively high emissions derived by the COMET model in north-western Poland / north-eastern Germany. Such differences on smaller spatial scales are probably partly due to differences in model transport and different weighting of the observations (i.e. different assumptions of model-data mismatch errors), but may reflect to some extent also some noise of the inverse modelling systems

[revised manuscript text omitted]
 | Model type | Meteorology | Background CH$_4$ (regional models) | Inversion technique |
|-------|-------------|-------------------------------------------------------|----------|------------|-------------|--------------------------------------|---------------------|
| TM5-4DVAR | EC JRC | Europe: 1° × 1° Global: 6° × 4° | 25 | Eulerian | ECMWF ERA-INTERIM | | 4DVAR |
| TM5-CTE | FMI | Europe: 1° × 1° Global: 6° × 4° | 25 | Eulerian | ECMWF ERA-INTERIM | | EnKF |
| TM3-STILT | MPI-BGC | Europe: 0.25° × 0.25° (STILT) Global: 5° × 4° (TM3) | 61 (STILT) 26 (TM3) | Lagrangian (STILT) Eulerian (TM3) | ECMWF operational analysis (STILT) ECMWF ERA-INTERIM (TM3) | TM3[5] | 4DVAR |
| LMDZ | LSCE | Europe: ~1.2° × 0.8° Global: ~ 7° × 3.6° | 19 | Eulerian | Nudged to ECMWF ERA-INTERIM | | 4DVAR |
| NAME | Met Office | 0.5625° × 0.375° [1] 0.3516° × 0.2344° [2] | 31[3] 59[4] | Lagrangian | Met Office Unified Model (UM) | based on meas. at Mace Head[6] | simulated annealing |
| CHIMERE | LSCE | 0.5° × 0.5° | 29 | Eulerian | ECMWF ERA-INTERIM | LMDZ[6] | analytical |
| COMET | ECN | 0.17° × 0.17° | 60 | Lagrangian | ECMWF ERA-INTERIM | TM5-4DVAR | analytical |

[1] for simulation period 01/2006-03/2010

[2] for simulation period 03/2010-12/2012

[3] for simulation period 01/2006-10/2009

[4] for simulation period 10/2009-12/2012

[5] coupling based on the method of *Rödenbeck et al.* [2009],

[6] further optimized in the inversion

[revised manuscript text omitted]